# Structural-functional brain network coupling during cognitive demand reveals intelligence-relevant communication strategies

Johanna L. Popp [1] ✉, Jonas A. Thiele [1], Joshua Faskowitz[2], Caio Seguin[2,3], Olaf Sporns [2] & Kirsten Hilger [1] ✉

Intelligence is a broad mental capability influencing human performance across tasks. Individual differences in intelligence have been linked to characteristics of structural and functional brain networks. Here, we consider their alignment, the structural-functional brain network coupling (SC-FC coupling) during resting state and during active cognition, to predict general intelligence. Using diffusion-weighted and functional magnetic resonance imaging data from 764 participants of the Human Connectome Project (replication: $N_1 = 126$, $N_2 = 180$), we model SC-FC coupling with similarity and communication measures that capture functional interactions unfolding on top of structural brain networks. By accounting for variations in brain region-specific neural signaling strategies, we show that individual differences in SC-FC coupling patterns predict individual intelligence scores. Most robust predictions result from cognitively demanding tasks and task combinations. Our study suggests the existence of an intrinsic SC-FC coupling organization enabling fine-drawn intelligence-relevant adaptations that support efficient information processing by facilitating brain region-specific adjustment to external task demands.

Intelligence describes the capacity to reason, plan, solve problems, think abstractly, comprehend complex ideas, and learn from experience[1]. Based on the observation that people who perform well on one type of cognitive task also tend to excel at other tasks, Spearman proposed the concept of general intelligence (*g*), influencing performance across all cognitive tasks and explaining individual variations[2]. Established tests to measure general intelligence were developed and enabled the discovery of its substantial impact on academic and professional accomplishments[3,4], socioeconomic status[5], and longevity[6], ultimately shaping life trajectories.

Individual differences in general intelligence have been associated with brain structure[7,8] and brain function[9,10] (for review see Hilger et al.[11]). Dominant neuro-cognitive models of intelligence, such as the Parieto-Frontal Integration Theory[12] and the Multiple Demand System[13], suggest a key role of frontal and parietal regions along with additional involvement of brain regions in temporal and occipital cortices. A more recent meta-analysis confirmed the importance of these regions, while also proposing critical relevance of the insular cortex, posterior cingulate cortex, and

subcortical structures[14]. Further, the Neural Efficiency Hypothesis initially suggested that more intelligent individuals exhibit lower, thus more efficient, activation of predominantly frontal brain regions during cognitive tasks[15]. However, subsequent reevaluation identified moderating variables such as sex, task type, task complexity, and brain area[16]. Finally, most recent proposals subsume evolving evidence from network neuroscience studies and conclude that distinct attributes of structural and functional brain networks are critical for understanding individual differences in general intelligence[17,18].

While structural brain networks approximate physical connections between brain regions (axons and dendrites), functional brain networks are constructed based on statistical relationships between time courses of neural activation[19]. Characteristics of both structural connectivity (SC) and functional connectivity (FC) have been related to intelligence[20–24] and even allow for significant prediction of individual intelligence scores in cross-validated predictive modeling approaches[25,26] (for review see Hilger et al.[11]; Hilger and Sporns[18]).

[1]Department of Psychology I, Würzburg University, Würzburg, Germany. [2]Department of Psychological and Brain Sciences, Indiana University, Bloomington, IN, USA. [3]Department of Psychiatry, The University of Melbourne, Parkville, VIC, Australia. ✉e-mail: johanna.popp@uni-wuerzburg.de; kirsten.hilger@uni-wuerzburg.de

However, although the best predictions of intelligence were achieved by system-wide approaches integrating multiple neuroimaging modalities[27-30], the potential to predict intelligence from variables unifying different modalities before entering a statistical model (i.e., feature-level multi-modality) remains widely unknown. Here, we assess the alignment between structural and functional brain networks, referred to as SC–FC coupling. Combining SC and FC prior to statistical analyses offers key advantages, as any reduction in the number of variables within a statistical model lowers the danger of overfitting and improves computational efficiency[31]. Most importantly, by assessing the alignment of structural and functional brain networks, we create a biologically meaningful measure that allows for insights concerning the interplay between structural pathways and functional communication[32].

Individual differences in SC–FC coupling have been linked to variations in specific cognitive traits[33-35], also suggesting their potential as a marker of general intelligence. While complex operationalizations of SC–FC coupling, such as biophysical models[36,37] are computationally costly, simple statistical (correlative) approaches are limited concerning mechanistic insight and biological plausibility[38]. Operationalizing SC–FC coupling with similarity and communication measures balances both approaches and thus holds potential for studying individual differences[32,39,40] where large sample sizes are required[41,42].

Both similarity and communication measures are computed based on SC and then compared to FC, thus defining SC–FC coupling. Similarity measures simply depict the resemblance of regional SC profiles to one another. Therefore, their comparison to FC serves as baseline for SC–FC coupling. In contrast, communication measures approximate FC on the basis of SC by quantifying the ease of communication between brain regions under an assumed model of network communication, spanning from diffusion processes to complex routing protocols[37,39,43,44]. The similarity of communication measures to the empirical FC is suggested to carry information about neural signaling processes[44,45], further elucidating the biological bases of individual differences in behavior. Strikingly, only SC–FC coupling during the task-free resting state (intrinsic SC–FC coupling) has been extensively studied[32,38,40,46], while SC–FC coupling during active cognition has neither been comprehensively characterized nor set in relation to behavioral phenotypes.

Even though task demands align FC profiles across participants, the remaining between-participant variance is suggested to be more stable and trait-like[47,48], and especially trait-relevant tasks (e.g., cognitive tasks for studying cognitive ability) were observed to facilitate the detection of meaningful brain–behavior associations[27,49-51]. For investigating SC–FC coupling, the consideration of task-induced FC might thus be particularly fruitful, as strong structural communication pathways may exist that are not functionally used during resting state but might play an important role in

supporting inter-regional interactions during cognitive demand. However, whether this influences SC–FC coupling strength and ameliorates relationships with intelligence remains an open question.

To address these gaps, we used data from 764 participants of the Human Connectome Project (HCP)[52,53] to investigate SC–FC coupling operationalized with one similarity and three communication measures, capturing the range of signaling strategies from routing to diffusion[39,45], during different (task) conditions. We first characterized task-induced SC–FC coupling on a brain-average and brain region-specific level and then contrasted it with intrinsic (resting-state) coupling. This revealed an intrinsic SC–FC coupling organization that enables fine-drawn, yet intelligence-relevant adaptations to specific tasks. Next, the relationship between general intelligence and brain-average SC–FC coupling was examined with a correlative approach, and a cross-validated predictive modeling framework was developed to study brain region-specific associations. Prediction performances were significant, with most robust predictions from cognitively demanding tasks and task combinations. These findings suggest that adaptations in the SC–FC coupling facilitate flexible adjustment to external task demands, thereby supporting efficient, intelligence-relevant information processing. All analyses were repeated in a lockbox sample from the HCP and two completely independent samples from the Amsterdam Open MRI collection (AOMIC)[54-56].

## Results
### General intelligence

General intelligence was operationalized as latent *g*-factor from 12 cognitive measures (Supplementary Table S1) using bi-factor analysis[25,57]. Details about the model, including factor loadings of each cognitive measure onto the *g*-factor, are reported in Supplementary Fig. S1. The data of the HCP[52,53] was divided into a main sample ($N = 532$) and a smaller subsample for later lockbox replication ($N = 232$). Scores in the main sample ranged between $-2.38$ and $2.40$ ($M = 0.25$; $SD = 0.83$), and their frequency distribution is visualized in Supplementary Fig. S2a, suggesting normality.

### Descriptive characterization of structural-functional brain network coupling

**Highest structural-functional brain network coupling during resting state and when operationalized with cosine similarity.** To operationalize structural-functional brain network coupling (SC–FC coupling), structural brain networks were constructed from diffusion-weighted imaging, while functional brain networks were derived from eight fMRI conditions (Table 1) using a multimodal parcellation scheme (358 nodes[58]). Individual structural brain networks were transformed into one similarity and three communication measures (Table 2) before computing region-specific SC–FC coupling values, defined as the Pearson correlation between regional connectivity profiles of the similarity or

## Table 1 | FMRI conditions

| | FMRI condition | Description | Frames per run | Run duration (min:sec) |
|---|---|---|---|---|
| 1 | Resting state (RES) | Participants' eyes are open and focused on a fixation cross. | 1200 | 14:33 |
| 2 | Emotion processing (EMO) | Matching of angry and fearful faces or of different shapes. | 2 × 176 | 2:16 |
| 3 | Gambling (GAM) | Card guessing game with the chance to win or lose money. | 2 × 253 | 3:12 |
| 4 | Language (LAN) | Auditory presentation of a brief story followed by questions about the topic of the story. | 2 × 316 | 3:57 |
| 5 | Motor (MOT) | Visual cues asking the participant to tap their fingers, squeeze their toes, or move their tongue. | 2 × 284 | 3:34 |
| 6 | Relational processing (REL) | Presentation of two pairs of objects and subsequent decision whether the objects have similar shape or texture. | 2 × 232 | 2:56 |
| 7 | Social cognition (SOC) | Presentation of short video clips with interacting vs. randomly moving objects (squares, circles, triangles). Deciding whether a social interaction had occurred or not. | 2 × 274 | 3:27 |
| 8 | Working memory (WM) | 2-back and 0-back working memory task, including faces, places, tools, and body parts. | 2 × 405 | 5:01 |

FMRI data were acquired in two, 2-hour long sessions during the resting state and seven tasks. Besides scan length, the image acquisition parameters did not differ between the eight fMRI conditions[71].

## Table 2 | Measures used to operationalize structural-functional brain network coupling

| | Description | Reference |
|---|---|---|
| **Similarity measure** | | |
| Cosine similarity (CoS) | Similarity of structural connectivity profiles based on vector orientation. | 110 |
| **Network communication measure** | | |
| Path length (PL)* | Length of the shortest available path between respective brain regions. | 43 |
| Communicability (G) | Signal transmission via diffusive broadcasting process. In a network, all walks (sequence of traversed edges) of all lengths are taken into account, and the contribution of each walk is inversely proportional to its length. | 111 |
| Search information (SI)** | Likelihood for a random walker to travel between two nodes via their shortest path. | 112 |

Measures whose matrices were adjusted after computation are marked with asterisks (* = matrices were inverted; ** = matrices were symmetrized and inverted, see Supplementary Information pp. 46–48; Table adapted from Popp et al.[32]).

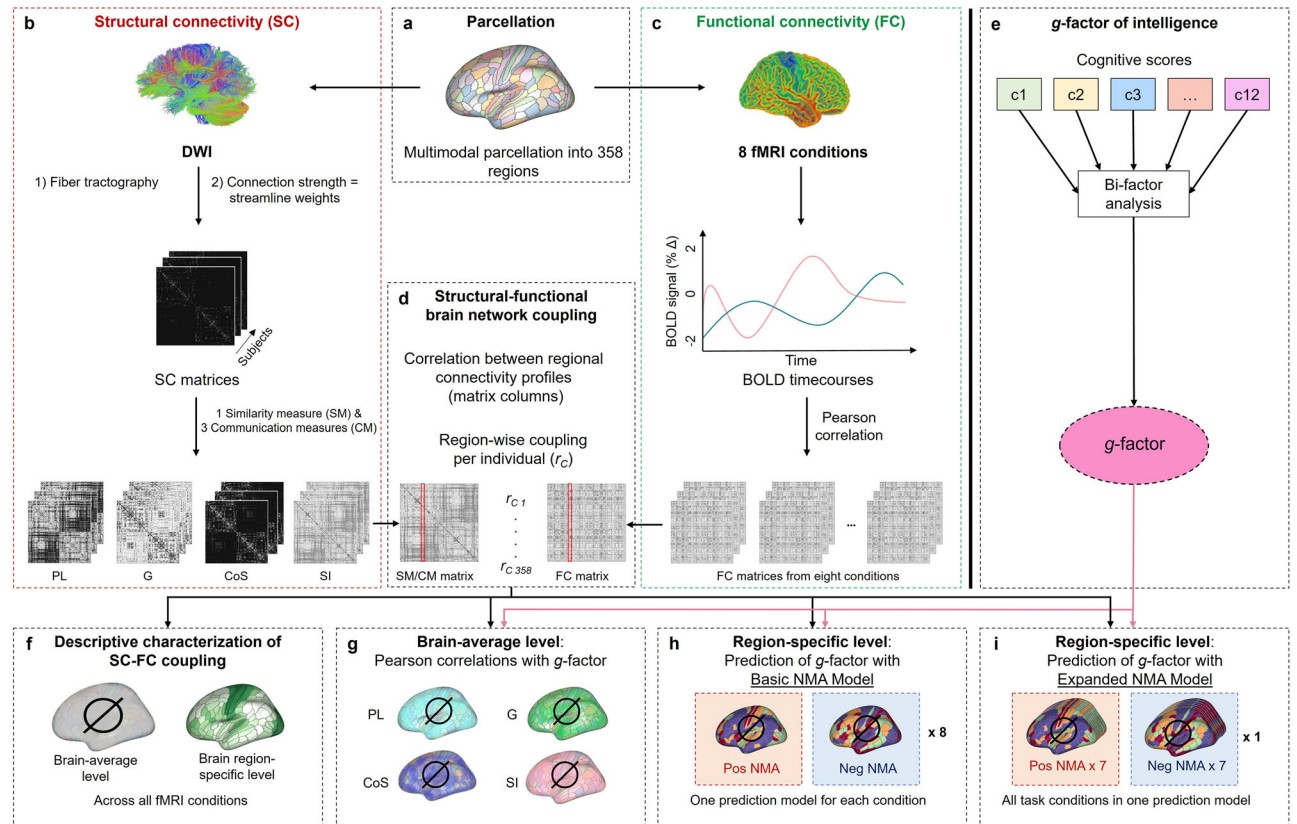

**Fig. 1 | Schematic overview of study procedures. a** Based on the multimodal parcellation of Glasser et al.[58], DWI and fMRI data were parcellated into 358 brain regions. **b** Individual SC matrices were transformed into one similarity measure (SM) and three communication measures (CM) expressing similarity in regional SC (CoS) or approximating regional FC under an assumed model of signal propagation based on the underlying SC (PL, G, SI), respectively (Table 2). **c** FC matrices reflect Pearson correlations between all possible pairs of regional blood oxygen level-dependent time courses for each of the eight fMRI conditions (Table 1). **d** The similarity matrix and the three communication matrices of each subject were compared to the condition-specific matrix of actual FC from each subject (4 × 8 comparisons) by correlating regional connectivity profiles (matrix columns) of the respective similarity or communication matrix with the corresponding regional connectivity profile in the condition-specific FC matrix. Each of the four comparisons per fMRI condition resulted in 358 individual coupling values ($r_C$, one per brain region). **e** Computation of latent g-factor from 12 cognitive scores using bi-factor analysis. **f** Descriptive characterization of general (group-average) SC–FC coupling on a brain-average and brain region-specific level. **g** Relationship between general intelligence and SC–FC coupling on brain-average level. **h** Prediction of general intelligence from region-specific SC–FC coupling with a predictive modeling framework using data from one fMRI condition at a time (Basic NMA Model; Supplementary Fig. S13). **i** Prediction of general intelligence from region-specific SC–FC coupling with data from all seven task fMRI conditions at a time (Expanded NMA Model; Supplementary Fig. S14). SC structural brain network connectivity, FC functional brain network connectivity, BOLD blood oxygen level dependent, SM similarity measure, CM communication measure, PL path length, G communicability, CoS cosine similarity, SI search information, NMA node-measure assignment.

communication networks and the individual condition-specific functional networks. Please note that a schematic overview of the complete study procedure is presented in Fig. 1. Across participants, brain-average SC–FC coupling strengths, obtained by averaging individual region-specific coupling values across all brain regions, differed significantly between fMRI conditions (repeated measures analysis of variance

(ANOVA): $F(7,3717) = [2091.1]$, $p < 0.001$): Compared to all other conditions, it was significantly higher during resting state and significantly lower during the emotion processing task (post-hoc pairwise comparisons: all $p < 0.05$; Supplementary Table S2; Fig. 2a). Similarly, coupling measures differed significantly in their brain-average coupling strengths (repeated measures ANOVA: $F(3,1539) = [9606.3]$, $p < 0.001$).

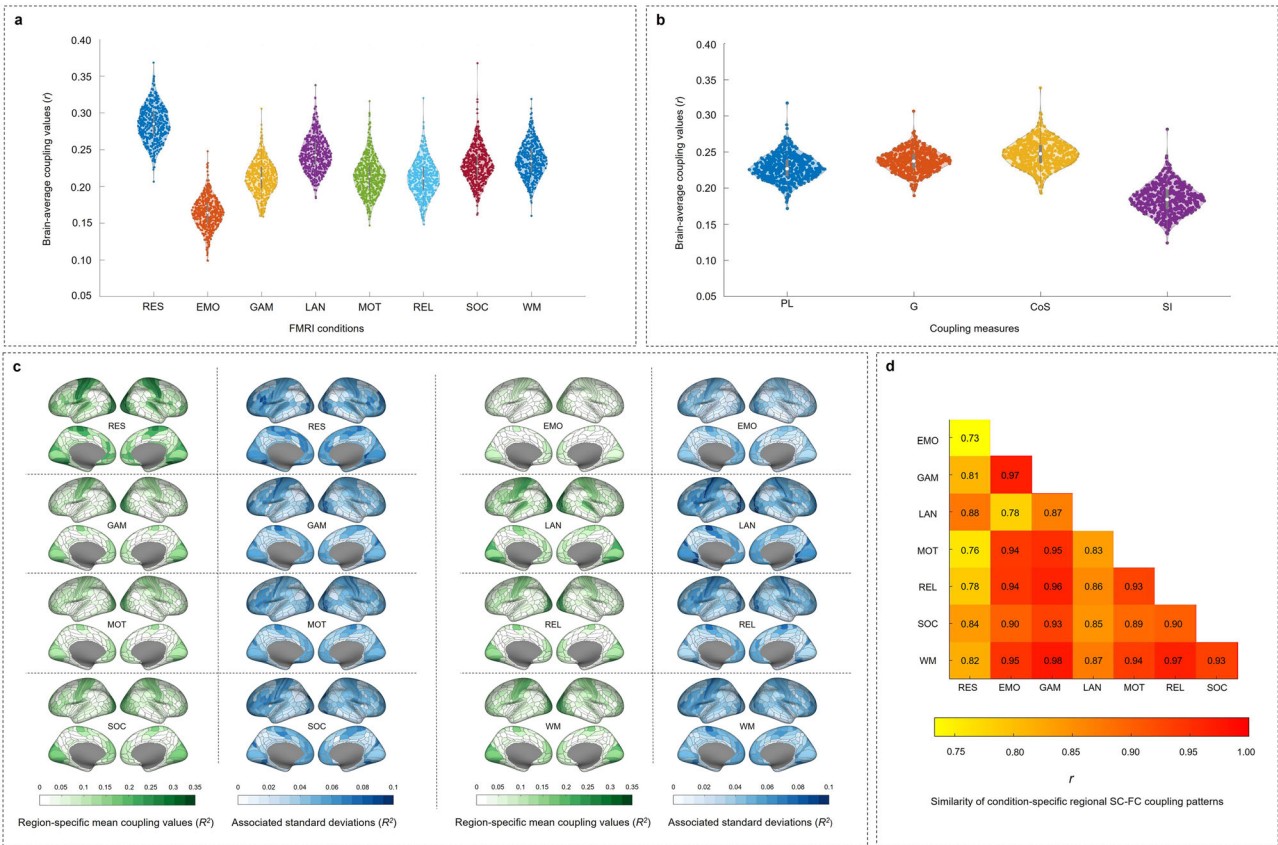

**Fig. 2 | Structural-functional brain network coupling varies between measures, fMRI conditions, and participants in the main sample ($N = 532$). a** Distribution of individual condition-specific brain-average SC–FC coupling values. **b** Distribution of individual measure-specific brain-average SC–FC coupling values. **c** Condition-specific group-average pattern of SC–FC coupling strength (in green). Blue maps indicate the variance in SC–FC coupling strength (as *SD*) across participants. **d** Heatmap depicting the similarity of group-average patterns of brain region-specific SC–FC coupling strength between eight fMRI conditions: Pearson correlation coefficients between condition-specific vectors representing the group-average pattern of SC–FC coupling strength. Note that significant group differences in (**a**) and (**b**) are reported in Supplementary Table S2 and Supplementary Table S3, respectively. RES resting state, EMO emotion processing task, GAM gambling task, LAN language task, MOT motor task, REL relational processing task, SOC social cognition task, WM working memory task, PL path length, G communicability, CoS cosine similarity, SI search information.

Cosine similarity provided significantly higher and search information significantly lower brain-average SC–FC coupling compared to all other coupling measures (post-hoc pairwise comparisons: all $p < 0.05$; Supplementary Table S3; Fig. 2b).

**Similar region-specific structural-functional brain network coupling patterns across tasks.** The group-average distribution of SC–FC coupling strength across the cortex during resting state closely matched the characteristic pattern reported in previous studies. This entails high coupling in unimodal areas and low coupling in multimodal areas[32,38,40,46] (Fig. 2c). Extending prior research, we revealed that SC–FC coupling patterns from different tasks were remarkably similar (all $r > 0.75$; mean $r = 0.91$), and the intrinsic SC–FC coupling pattern differed significantly more from task-induced SC–FC coupling patterns (mean $r = 0.80$) compared to how the task-induced SC–FC coupling patterns differed among one another (z-test on Fisher z-transformed correlations; $p < 0.001$; Fig. 2d).

**Task demands induce fine-drawn adaptations from resting-state structural-functional brain network coupling.** Adaptations in SC–FC coupling between conditions were quantified as absolute differences between group-average region-specific coupling patterns (see "Methods"; Supplementary Fig. S3). Task-specific adaptations from intrinsic SC–FC coupling were largest for the emotion processing task (+52.13%) and smallest for the language task (−32.06%), while adaptations from task-general SC–FC coupling were again largest for the emotion processing task (+96.49%) but smallest for the gambling task (−54.68%; Supplementary Fig. S3). In both cases, adaptations were most pronounced in unimodal brain areas (Supplementary Fig. S4).

**Structural-functional brain network coupling is associated with general intelligence**
General intelligence was significantly associated with brain-average SC–FC coupling during the emotion processing task. Significant partial correlations controlling for effects of age, gender, handedness, and in-scanner motion were consistently observed across all coupling measures (Supplementary Table S4). SC–FC coupling during all other fMRI conditions was not associated with general intelligence (all $r \leq 0.11$; all $p > 0.0125$).

**Structural-functional brain network coupling predicts individual intelligence scores across different conditions**
The Basic Node-Measure Assignment Model (B-NMA) is a 5-fold internally cross-validated prediction framework developed to predict individual differences in a phenotypical variable from SC–FC coupling[32]. It is based on multiple linear regression and creates prediction features by assigning coupling measures to brain regions based on their association with the phenotypical variable of interest. Significant prediction of individual intelligence scores from brain region-specific SC–FC coupling was possible with all fMRI conditions except the motor task (Table 3). Across-task differences in prediction performance reached significance for the social cognition task and the working memory task,

**Table 3 | Performance when predicting intelligence from brain region-specific structural-functional brain network coupling**

| FMRI condition | Prediction performance | | |
|---|---|---|---|
| | r | R² | p |
| Resting state (RES) | 0.18 | 0.03 | 0.002* |
| Emotion processing (EMO) | 0.12 | 0.01 | 0.029* |
| Gambling (GAM) | 0.18 | 0.03 | <0.001* |
| Language (LAN) | 0.15 | 0.02 | 0.001* |
| Motor (MOT) | 0.07 | <0.01 | 0.072 |
| Relational processing (REL) | 0.12 | 0.01 | 0.014* |
| Social cognition (SOC) | 0.22 | 0.05 | <0.001* |
| Working memory (WM) | 0.25 | 0.06 | <0.001* |
| All tasks (ALL) | 0.27 | 0.07 | <0.001* |

The Basic NMA Model was used to predict intelligence based on brain region-specific coupling obtained during single tasks, while the Expanded NMA Model was used to predict intelligence from task-combined region-specific coupling (see "Methods"). P values indicating significant associations are marked with an asterisk (*$p < 0.05$, non-parametric permutation tests).

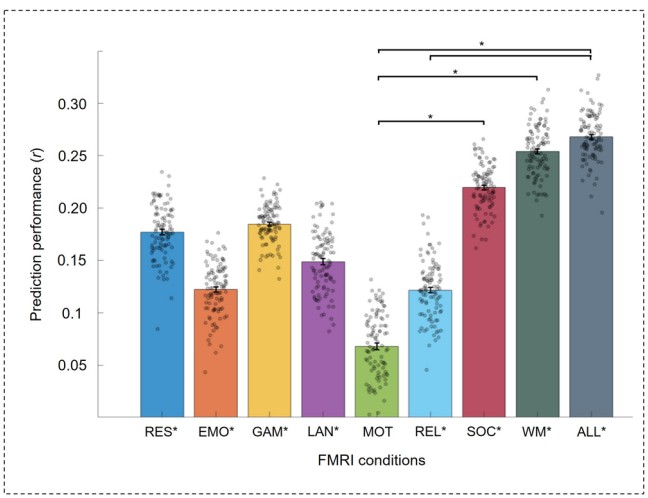

**Fig. 3 | Performance to predict intelligence from structural-functional brain network coupling during different fMRI conditions (main sample, $N = 532$).** Predictions were cross-validated (5-fold) and are based on brain region-specific coupling from eight separate fMRI conditions (Basic NMA Model) as well as from task-combined coupling (Expanded NMA Model). Bar graphs depict mean prediction values across 100 repetitions with different training-test splits, and error bars indicate the standard error of the mean. Significance of prediction performance was assessed with non-parametric permutation tests, while significant differences in prediction performance were evaluated with model difference tests (see "Methods"). P values indicating significant predictions are marked with an asterisk a) below bar graphs for single models and b) above bar graphs for between-model differences (*$p < 0.05$). RES resting state, EMO emotion processing task, GAM gambling task, LAN language task, MOT motor task, REL relational processing task, SOC social cognition task, WM working memory task, ALL all task fMRI conditions.

which predicted significantly better than the motor task ($p < 0.05$; Supplementary Table S5; Fig. 3). The condition-specific intelligence-associated communication strategies, defined in the framework as positive and negative node-measure assignments (NMAs, see "Methods"), are illustrated in Fig. 4.

### Best prediction when combining coupling information across tasks
The Expanded Node-Measure Assignment Model (E-NMA) extends the Basic NMA Model by combining brain region-specific coupling information

across all tasks. Again, significant prediction of intelligence was achieved ($r = 0.27$, $R^2 = 0.07$, $p < 0.001$; Table 3). This prediction was significantly better than predictions with the Basic NMA Model from the motor task and the relational processing task (all $p < 0.05$; Supplementary Table S5; Fig. 3).

### Replication
To evaluate the robustness and generalizability of our findings, we conducted the same analyses (replication) in a lockbox sample from the HCP ($N = 232$)[52,53] and in two completely independent samples from the AOMIC (AOMIC PIOP1, $N = 126$; AOMIC PIOP2, $N = 180$)[54–56].

**General intelligence.** Supplementary Fig. S2b–d illustrates the distributions of general intelligence scores in the lockbox sample (latent g-factor; range = −2.43 to 2.12; $M = 0.15$; $SD = 0.83$), and in the two independent samples where it was operationalized as sum score of the Raven's Advanced Progressive Matrices (RAPM) Test[59,60] (AOMIC PIOP1: range = 12–35; $M = 24.90$; $SD = 4.82$; AOMIC PIOP2: range = 13–34; $M = 24.68$; $SD = 4.82$).

**Descriptive characterization of structural-functional brain network coupling.** Similar to the main sample, also in the lockbox and both AOMIC samples, brain-average SC–FC coupling strengths differed significantly between fMRI conditions (Supplementary Tables S6–S8; Supplementary Figs. S5a–S7a). However, only in the lockbox sample but not in the AOMIC samples, coupling was significantly higher during resting state and significantly lower during the emotion processing task compared to all other conditions, like in the main sample (all $p < 0.05$; Supplementary Tables S6–S8).

Similar to the main sample, brain-average SC–FC coupling strength differed significantly between coupling measures in the lockbox and both AOMIC samples (Supplementary Tables S9–S11; Supplementary Figs. S5b–S7b). Significantly higher coupling when operationalized with cosine similarity compared to all other measures could be replicated in the lockbox and both AOMIC samples, while significantly lower coupling when operationalized with search information compared to all other measures was also replicated in all three samples (Supplementary Tables S9–S11).

The group-average cortical distribution of SC–FC coupling strengths for all fMRI conditions in the lockbox sample and both AOMIC samples aligned with the pattern observed in the main sample (Supplementary Figs. S5c–S7c). Also, the across-task similarity in SC–FC coupling patterns replicated in all three samples (all $r \geq 0.75$). Like in the main sample, the intrinsic SC–FC coupling pattern differed significantly more ($p < 0.001$) from task-induced patterns compared to task-induced patterns among one another in the lockbox sample (mean $r = 0.80$ vs. $r = 0.90$) and in the AOMIC PIOP2 sample (mean $r = 0.89$ vs. $r = 0.95$), but not in the AOMIC PIOP1 sample (mean $r = 0.93$ vs. 0.91).

**Structural-functional brain network coupling and general intelligence.** Similar to the main sample, also in the lockbox sample, brain-average SC–FC coupling during the emotion processing task was positively associated with general intelligence across all measures. However, associations were of smaller magnitude and did not reach significance in this smaller sample (all $p > 0.0125$; Supplementary Table S12). In the AOMIC samples, no significant associations between brain-average SC–FC coupling and general intelligence were observed (all $p > 0.0125$; Supplementary Tables S13 and S14).

The Basic NMA Model, applied in a 5-fold internal cross-validation, significantly predicted individual intelligence scores within the lockbox sample from the language task ($r = 0.25$; $p = 0.006$) and the working memory task ($r = 0.17$; $p = 0.018$; Supplementary Table S15; Supplementary Fig. S8). Also, the significant prediction from task-combined coupling information replicated in the lockbox sample ($r = 0.21$, $p = 0.003$; Supplementary Table S15). In the smaller AOMIC samples, intelligence predictions did not reach significance (all $r < 0.12$; all $p > 0.05$; Supplementary Tables S16 and S17; Supplementary Figs. S9 and S10).

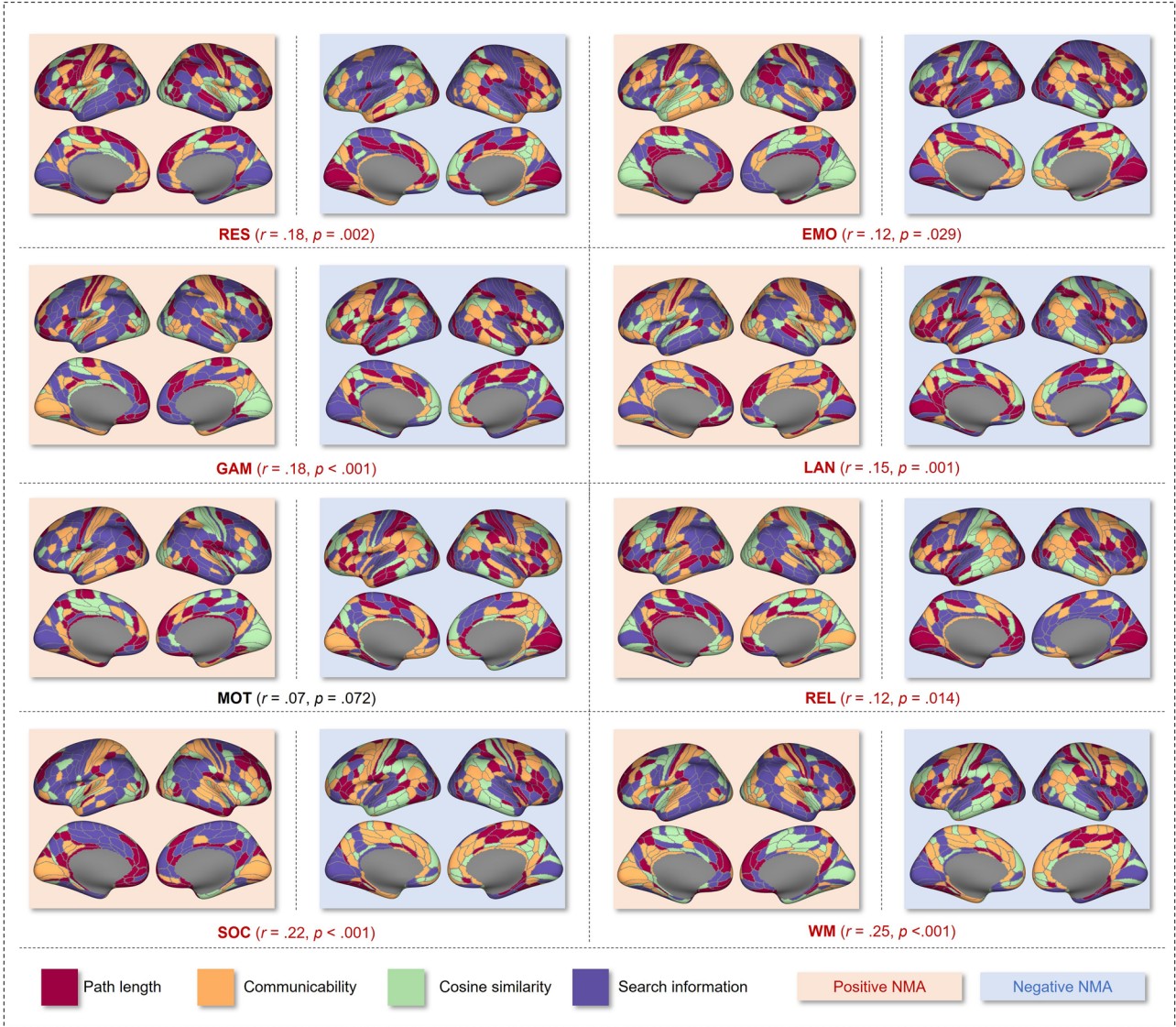

**Fig. 4 | Brain region-specific communication strategies predict general intelligence in the main sample (N = 532).** Assignments of coupling measures to brain regions (NMAs) were defined by the largest positive (red background) and negative (blue background) magnitude associations between coupling measures and general intelligence across participants. Assignments were created for each fMRI condition separately (Basic NMA Model). Associated prediction performance and significance are reported in parentheses. Note that the visualized NMAs in this figure are based on data from the whole main sample, whereas the NMAs applied in the prediction models were only derived from data of the training samples. RES resting state, EMO emotion processing task, GAM gambling task, LAN language task, MOT motor task, REL relational processing task, SOC social cognition task, WM working memory task.

The cross-sample model generalization test applying the Basic NMA Model with parameters derived from the main sample to the lockbox sample, was successful for resting state ($r = 0.15$; $p < 0.001$), language task ($r = 0.19$; $p < 0.001$), motor task ($r = 0.08$; $p < 0.001$), relational processing task ($r = 0.13$; $p = 0.002$), social cognition task ($r = 0.23$; $p < 0.001$), and working memory task ($r = 0.11$; $p = 0.031$; Supplementary Table S18). Prediction performance differed significantly between multiple prediction models (model difference test; Supplementary Table S19; Supplementary Fig. S11), with the motor task again performing significantly worse than the social cognition task ($p < 0.05$; Supplementary Table S19). Although AOMIC tasks differed from those of the HCP, the cross-sample model generalization tests were also successful for the working memory task ($r = 0.15$; $p < 0.001$) and the emotion processing/matching task ($r = 0.12$; $p < 0.001$; Supplementary Table S20) in the AOMIC PIOP1 sample and for resting state ($r = 0.05$; $p = 0.006$) and the emotion processing/matching task ($r = 0.04$; $p = 0.007$; Supplementary Table S21) in the AOMIC PIOP2 sample.

Finally, the cross-sample model generalization test applying the Expanded NMA Model with all 14 parameters of the main sample to the lockbox sample was successful ($r = 0.20$; $p < 0.001$; Supplementary Table S18). Similar to the main analysis, this prediction was significantly better than the prediction from the motor task ($p < 0.05$; Supplementary Table S19; Supplementary Fig. S11). Cross-sample model generalization tests of the Expanded NMA Model to the AOMIC samples were not possible due to fewer and differing task fMRI conditions.

**Post-hoc analyses**
As scan length varied across fMRI conditions (Table 1) and it has been shown that scan length could influence the reliability of FC estimates[61], we repeated our main analyses on fMRI data, trimming each scan to match the duration of the shortest one (176 frames per run; emotion processing task). Similar results were observed: Brain-average SC–FC coupling differed significantly between fMRI conditions (Supplementary Fig. S12) and was significantly higher during the resting state and significantly lower during the emotion processing task

compared to all other conditions (Supplementary Table S22). Further, the results of the intelligence predictions based on region-specific SC–FC coupling values were nearly identical (compare Table 3 with Supplementary Table S23).

To enhance conceptual insights, we post-hoc classified the tasks during fMRI recording into more or less cognitively demanding ones. Specifically, we determined the associations (Pearson correlations) between general intelligence and behavioral task performance across all tasks (as operationalized in Greene et al.[51]). This analysis was conducted with all HCP participants for whom the respective data were available ($N = 737$). Significant associations (Bonferroni corrected threshold for six correlations; sig. $p < 0.008$) were observed for the emotion processing task ($r = 0.13$; $p < 0.001$), the language task ($r = 0.45$; $p < 0.001$), the relational processing task ($r = 0.45$; $p < 0.001$), the social cognition task ($r = 0.23$; $p < 0.001$), and the working memory task ($r = 0.49$; $p < 0.001$). The language task, the relational processing task and the working memory task demonstrated the strongest associations ($r \geq 0.45$) and were thus considered as tasks of higher cognitive demand.

## Discussion

The present study investigates how the coupling between structural and functional brain networks varies with different cognitive demands, how it depends on different coupling measures, and whether it allows to predict individual general intelligence scores. We used fMRI data acquired during different conditions ($N_{main} = 532$; replication: $N_{lockbox} = 232$; $N_{PIOP1} = 126$; $N_{PIOP2} = 180$) and operationalized SC–FC coupling with one similarity and three communication measures. Regardless of the measure, coupling was highest during the resting state and lowest during emotion processing. Irrespective of the task, cosine similarity provided the highest and search information the lowest coupling. Across tasks and measures, the highest coupling was present in unimodal areas, while the lowest coupling was observed in multimodal areas. On a brain-average level, higher intelligence was only associated with higher coupling during the emotion processing task. However, accounting for region-specific variability in SC–FC coupling enabled significant prediction of individual intelligence scores, with cognitively demanding tasks and coupling information from all tasks providing strong and most robust (replicable) predictions.

The observation that coupling is highest during the resting state suggests that intrinsic connectivity more closely resembles anatomical brain connections (as uncovered with diffusion tractography) than task-induced connectivity. Assuming that SC may reflect the history of neural co-activation[62,63], and that SC–FC coupling represents how momentary co-activation patterns align with this history, it seems plausible that highest coupling occurs during the brain's default state, which serves as a common foundation for all task-induced adaptations to build upon and also dominates the individual pattern of FC in the presence of any external demand[64,65]. However, specific tasks may differ both in their extent of required adaptation from this default mode and in their frequency of consultation (their history). Thus, our result of lowest coupling during the emotion processing task may suggest that the specific demand imposed by this task is either recruited less often (and thus less reflected in SC) and/or requires stronger adaptations from the intrinsic FC pattern. Given that these findings could not be replicated in the independent AOMIC samples, further research is required to draw stronger conclusions.

The condition-consistent observation that when averaging coupling values across the whole brain, a measure simply depicting the similarity of regional SC better predicts FC than any of the communication measures, supports previous evidence revealing the existence of significant variation in preferred communication strategies across different brain regions[40,66]. Thus, no communication measure that assumes a specific signaling strategy can, on its own, sufficiently explain the actual FC. Similarity measures, in contrast, do not assume a specific strategy, but are simply based on the assumption that brain regions with similar SC profiles (i.e., anatomical inputs and outputs) have similar FC profiles. This straightforward idea seems to be more predictive than any of the communication models—at least when averaging across the whole brain.

The global pattern of regional SC–FC coupling strength (high coupling in unimodal areas; low coupling in multimodal areas), previously identified in resting-state studies[32,40,46,67], persists during varying cognitive demands. However, we also revealed fine-drawn adaptations between conditions, with larger differences between resting state and tasks than among different tasks. This observation aligns with previous FC studies[64,68,69], and further substantiates the assumption that the network architecture required to meet a particular cognitive demand is primarily determined by the intrinsic network architecture[70], and only secondarily modified through smaller task-general and task-specific adaptations[64].

On a brain-average level, functional interactions operate more closely along structural pathways in more intelligent individuals only during the emotion processing task, which is suggested to involve relatively low cognitive demand[71,72]. This was consistently observed across all coupling measures and may indicate an efficient use of the underlying structure without the need for more fine-drawn regional adaptations in the face of low demand (in line with Haier et al.[15] and Neubauer & Fink[15]). As this would help to save cognitive resources necessary for handling potential future higher demands, such a coupling organization could be interpreted as being adaptive, emphasizing that the ability for efficient adaptation presents a hallmark of human intelligence[73]. Since the replication of this finding was neither possible in the lockbox nor the independent AOMIC samples, our interpretation should be considered with caution.

On a brain region-specific level, our results reveal that variations in the preferred signaling strategy are critically involved in the prediction of intelligence, especially for tasks with higher cognitive load[27,71]. This supports the idea that efficient information processing could not be achieved when neural signaling is only dominated by one strategy, but that brain regions perform best with different strategies based on their individual function.

Notably, strong and most robust (replicable) prediction performance was achieved with coupling acquired during cognitively demanding tasks (i.e., working memory and the language task). This observation aligns with recent research[27,49,50,72,74] and is substantiated by established personality theories, which propose that individual variations in a given personality trait become most apparent during situations relevant for this trait[75–77]. Cognitively demanding tasks may rely more on intelligence-relevant neural circuity, emphasizing meaningful individual differences and thereby improving predictive accuracy and robustness (for similar argumentation see ref. [48]). However, prediction from resting state outperformed some of the task conditions. One potential explanation could be that the unconstrained resting state leads more intelligent individuals to follow more complex thoughts, thus unintentionally inducing trait-relevant situations.

Extending previous reports that demonstrate enhanced prediction of behavioral phenotypes when integrating FC across multiple tasks[27,78,79], combining individual coupling information across tasks descriptively yielded the best prediction performance in the internal cross-validation in the main sample. Note that this must not merely be an effect of including more factors in the statistical model, as more factors increase the likelihood for overfitting, which we prevent with cross-validation, and would therefore reduce prediction accuracy[80]. Regarding general intelligence, this observation is particularly interesting, as the *g*-factor theory proposes the existence of a latent intelligence factor that captures the common variance of an individual's performance across tasks[2]. Thus, neural processes that might reflect *g* at the brain level ("neuro-*g*"[81,82]) should also be involved in all tasks[83]. Our study provides initial evidence that task-combined brain region-specific structural-functional brain network coupling could be a potential candidate for such a proposed "neuro-*g*."

The vast majority of previous research predicting human phenotypes from multimodal neuroimaging data considered different brain modalities as separate prediction features, thus combining modalities at the level of the statistical model[28,29,84]. Our approach contrasts with this method as we combined brain modalities beforehand—at the feature level. This may present a promising route for future research, however, challenges that come with such an approach include additional researchers' degrees of freedom of how to integrate different modalities, increased computational resources[85],

the selection of a parcellation scheme that appropriately captures properties of both modalities, and reduced sample sizes as neuroimaging data of two or even more modalities must be of sufficient quality. Finally, and maybe most critically, the expected effect sizes of associations with individual differences in behavioral phenotypes are limited by the product of both modalities' reliabilities and thus are per se lower than when predicting from only one modality. Consequently, significant associations between behavioral phenotypes and multimodal brain features are harder to detect. Nevertheless, we consider multimodal feature prediction as a fruitful extension, as it can provide insights into human brain functioning that cannot be concluded from unimodal feature prediction—as long as these challenges are addressed appropriately. For instance, uncertainties about the best combination method can be overcome by appropriate control- or multiverse analyses[86], multimodal brain parcellations have been developed[58], and the problem of limited reliabilities could be addressed by enhancing statistical power through larger sample sizes or through increasing effect sizes by further methodological development[41,87,88]. Promising combinations for future research include positron emission tomography and FC, which could inform how the availability of certain proteins (i.e., transporters and channels) might relate to synchronous activity of brain regions linked to the behavior of interest.

There are limitations to this study. First, our samples were restricted to young adults, posing the question of result generalizability to a broader age range. Second, neuroimaging data are inherently noisy[88], and both FC and SC have limited reliability[89,90]. Although we applied state-of-the-art methods addressing these issues, the remaining influences of confounds cannot be ruled out. Third, our analyses focused on one similarity and three communication measures as well as on the application of two specific prediction frameworks (B-NMA, E-NMA), while additional measures[39,40,66] and alternative prediction frameworks (e.g., PLS-regression[27,91]; neural networks[92]) are available. Although the framework developed here is computationally very efficient and allows for intuitive interpretations, the systematic evaluation of our framework against alternative methods may offer further insights. Especially approaches combining different communication strategies[66] and methods that account for potential non-linear relations may present promising starting points for future research on the relationship between intelligence and SC–FC coupling. Fourth, task variety and difficulty were limited. As it has been shown that relations between intelligence and neural activation can differ depending on task content and difficulty[93,94], expanding both may provide complementary insights. Fifth, it is important to acknowledge that although reaching significance in the main sample, not all of our results replicated in the lockbox and the independent samples. Possible reasons include smaller sample sizes (lockbox and AOMIC samples) as well as differences in (a) scanner sites, (b) image acquisition parameters, (c) task conditions, and (d) data quality in the AOMIC samples. Finally, forthcoming studies could profit from (a) investigating the relationship between general intelligence and individual SC–FC coupling changes over longer time periods where SC underlies significant changes, (b) disentangling the individual contributions of SC and FC to the predictive power of SC–FC coupling measures, and (c) incorporating naturalistic in-scanner stimuli (e.g., movie paradigms), as these may posit a valuable compromise between overly constrained tasks lacking ecological validity and unrestricted resting state, thus potentially amplifying behaviorally relevant individual differences[47,48].

To conclude, our study reveals the presence of an intrinsic structural-functional brain network coupling organization that enables fine-drawn, but intelligence-relevant, task-dependent adaptations. We propose that these adaptations support efficient information processing by facilitating brain region-specific adjustment to external task demands. Finally, achieving the greatest prediction performance through combining system-wide coupling information from multiple tasks contributes to established intelligence theories that advocate the existence of a common intelligence factor on the neural level[81,82] and consider intelligence as an emergent whole-brain phenomenon[12–14,18]. Overall, our study exemplifies how combining different neuroimaging modalities within multimodal prediction features can further improve our understanding of the human brain.

## Methods

### Preregistration
The purpose of the study, the planned analyses, as well as all methodological details, were preregistered in the Open Science Framework: https://osf.io/ctvf9. Note that because this study was part of a larger project examining the relationship between SC–FC coupling and personality traits, study plans concerning general intelligence were listed as secondary analyses (see: "*Other*", *Subsection I*).

### Statistics and reproducibility
Detailed information about the sample and statistical analyses is provided in the subsequent sections to support reproducibility.

### Participants
Main analyses were performed on the HCP Young Adult Sample S1200[52,53], including 1200 participants (656 female; 1089 right-handed; mean age = 28.8 years; age range = 22–37 years).

Participants with (a) missing DWI, resting-state fMRI, or task fMRI, (b) missing personality scores or missing cognitive measures from 12 tasks required to compute a latent intelligence factor, or (c) a Mini-Mental State Examination score smaller than 27 (serious cognitive impairment) were excluded. According to Parkes et al.[95], participants with excessive in-scanner head motion measured by framewise displacement[96] during any fMRI condition were also excluded: mean framewise displacement > 0.20 mm; proportion of motion spikes (framewise displacement > 0.25 mm) > 20 percent; any motion spikes > 5.00 mm. The final sample comprised 764 participants (402 female; 697 right-handed; mean age = 28.6 years; age range = 22–36 years). Analyses were performed on 70% of this sample (main sample; HCP532; N = 532; 271 female; 485 right-handed; mean age = 28.6 years; age range = 22–36 years), while 30% were set aside for replication (lockbox sample; HCP232; N = 232; 131 female; 212 right-handed; mean age = 28.5 years; age range = 22–36 years).

### General intelligence
General intelligence was computed as latent *g*-factor from 12 cognitive measures[32,65] (Supplementary Table S1) with bi-factor analysis[25,57] using data from 1186 participants of the HCP[52,53]. This approach is standard in intelligence research[25,65,92] and has been shown to offer a better fit to diverse cognitive test scores than alternative models[97,98].

### Neuroimaging data acquisition and preprocessing
MRI scans were obtained on a Siemens Skyra 3T scanner equipped with a 32-channel head coil.

**Diffusion-weighted imaging (DWI).** SC was estimated based on minimally preprocessed DWI data (time to repetition = 5520 milliseconds (ms); time to echo = 89.5 ms; 1.25 mm isotropic voxel resolution; multiband acceleration factor = 3; $b$ = 1000, 2000, 3000 s/mm$^2$; 90 directions/shell)[52,99]. Bias correction, modeling of white matter fibers via constrained spherical deconvolution[100], and tissue normalization[101] were performed according to the MRTrix pipeline[102,103]. Finally, probabilistic streamline tractography was employed[104] and only streamlines fitting the estimated white matter orientations from the diffusion image were kept[99,105,106].

**Functional magnetic resonance imaging (fMRI).** FMRI scans were obtained using a gradient-echo EPI sequence in two sessions with multi-slice acceleration (time to repetition = 720 ms; time to echo = 33.1 ms; 2-mm isotropic voxel resolution; flip angle = 52°; multiband acceleration factor = 8)[52]. Both sessions included two 15-min resting-state scans in opposite phase encoding directions (L/R and R/L), followed by ~30 min of task fMRI comprising seven tasks (Table 1; each L/R and R/L) that were

split between both sessions. FMRI data were downloaded in the minimally preprocessed form. Additionally, 24 head motion parameters, eight mean signals from white matter and cerebrospinal fluid, and four global signals were regressed out according to Parkes et al.[95] (strategy No. 6). Task-evoked neural activation was removed by simultaneously adding basis-set task regressors. This method has been shown to be similarly effective as concurrent methods but achieves this performance with fewer parameters in the nuisance regression step, preserving more temporal degrees of freedom, and reducing the risk of overfitting[107].

### Structural and functional brain network connectivity
The multimodal parcellation scheme of Glasser et al.[58], dividing the cortex into 360 regions, was used to construct structural and functional brain networks. The left and right hippocampi were excluded due to their classification as subcortical during preprocessing, yielding a total of 358 brain regions. Individual-specific SC matrices entail the SIFT2 streamline density weights between all possible pairs of brain regions, while individual FC matrices for each of the eight fMRI conditions were constructed based on Fisher $z$-transformed Pearson correlations of regional blood oxygen level-dependent signal time courses between all possible pairs of brain regions. Note that SC matrices were modeled as symmetric, and FC matrices were first computed separately for the two different phase encoding directions (L/R and R/L) and subsequently averaged condition-wise[32,64,108].

### Structural-functional brain network coupling
Based on each individual's SC matrix, one similarity measure (cosine similarity, CoS) and three communication measures (path length, PL; communicability, G; search information, SI) were computed[32,40] (Table 2; details in Supplementary Information pp. 46–48). These measures were selected from multiple alternatives[32], as they are representative of four signaling categories (routing, path accessibility, diffusion, and similarity[39,45]). Brain region-specific coupling values were then obtained by correlating regional connectivity profiles (matrix columns) of each similarity or communication matrix with the respective regional connectivity profile of a condition-specific FC matrix ($4 \times 8$ comparisons). This resulted in 358 region-specific coupling values ($r_C$) for each comparison (for each fMRI condition and each measure; Fig. 1a–d) per participant.

### Descriptive characterization of structural-functional brain network coupling
**Brain-average level**. For each participant and each of the eight fMRI conditions, brain-average coupling values were computed as the mean over 358 region-specific coupling values ($r_C$). This was conducted for each of the four coupling measures separately (CoS, PL, G, SI). To examine coupling differences between fMRI conditions, these participant-specific brain-average coupling values were then averaged across measures. To assess differences between coupling measures, the participant-specific brain-average coupling values were averaged across fMRI conditions. The resulting values were visualized in violin plots, and significant differences in group means were examined with a repeated measures ANOVA and post-hoc pairwise comparisons using model-estimated marginal means.

**Brain region-specific level**. To analyze differences in the regional pattern of SC–FC coupling between fMRI conditions, we initially identified for each condition separately which similarity or communication measure per brain region was able to explain the highest amount of variance ($R^2$) in FC across all participants. This condition-specific group-general assignment between brain region and coupling measure was subsequently used to extract individual region-specific coupling values ($R^2$). These were then averaged across all participants, yielding a vector with 358 elements, which presents the basis for group-average brain maps illustrating the regional pattern of SC–FC coupling per fMRI condition. Statistical comparisons between group-average condition-specific maps were performed by computing Pearson correlations between those

condition-specific vectors for all possible pairs of fMRI conditions. Adaptations in SC–FC coupling from the resting state can be subdivided into task-general adaptations common to all tasks and task-specific adaptations unique to a certain task. Task-specific adaptations from intrinsic SC–FC coupling were quantified as the absolute difference between intrinsic and task-induced group-average region-specific coupling values, separately computed for each condition. Task-general adaptations were assessed as the absolute difference between intrinsic group-average region-specific coupling values and the mean over all task-induced group-average region-specific coupling values. Additionally, task-specific adaptations from a task-general pattern of SC–FC coupling strength (mean over all task-induced group-average region-specific coupling values from the remaining tasks) were computed. Note that this part of the analysis was not preregistered.

### Structural-functional brain network coupling and general intelligence
**Association between brain-average structural-functional brain network coupling and general intelligence**. Partial correlations between general intelligence scores and individual condition-specific brain-average coupling values were computed while simultaneously controlling for effects of age, gender, handedness, and in-scanner motion (condition-specific mean framewise displacement). This was conducted for each coupling measure separately. Significant associations were determined at $p < 0.05$. To account for multiple comparisons (four coupling measures), we applied Bonferroni correction ($p < 0.0125$).

**Prediction of general intelligence from brain region-specific structural-functional brain network coupling**
Basic node-measure assignment model (B-NMA). To investigate whether intelligence can be predicted from brain region-specific SC–FC coupling of a particular fMRI condition, a recently developed prediction framework was applied. Specifically, we used the B-NMA, which is based on multiple linear regression and was specifically developed for the prediction of phenotypical variables from structural-functional brain network coupling[32]. This framework explicitly considers region-specific variations in SC–FC coupling measures and was run for each of the eight fMRI conditions. In brief, for each coupling measure separately, region-specific coupling values ($r_C$) from participants of a training sample were correlated with general intelligence scores (partial correlations controlling for effects of age, gender, handedness, and in-scanner motion). This resulted in four correlation coefficients ($r_P$) per brain region. Two group-general NMA masks were then constructed by assigning the coupling measure with the largest positive magnitude association (positive NMA) and the coupling measure with the largest negative magnitude association (negative NMA) between intelligence and coupling strength to a given region. These masks were finally used to extract individual region-specific coupling values ($r_C$) from which two model input features were computed, one as the average across all individuals' coupling values extracted with the positive NMA and one as the average across all individuals' coupling values extracted with the negative NMA. Thus, the multiple linear regression model ($y = \beta_0 + \beta_1 X_1 + \beta_2 X_2 + \in$) included two predictors: (a) brain-average coupling values from individual positive NMAs ($X_1$), and brain-average coupling values from individual negative NMAs ($X_2$), where $\beta_0$ is the intercept, $\beta_n$ the regression coefficient, $\in$ the error term, and $y$ the predicted intelligence scores. General intelligence scores were predicted for each participant by using a 5-fold cross-validation scheme. Note that herein, NMAs built in the training sample were always used to extract region-specific coupling values in the test sample (Supplementary Fig. S13).

Expanded node-measure assignment model (E-NMA). To investigate whether the prediction of general intelligence can be enhanced when combining region-specific SC–FC coupling information across tasks, the "Expanded Node-Measure Assignment Model (E-NMA)" was developed. Again, two model input features were created for each fMRI condition

(excluding resting state), but these 14 condition-specific features were now considered simultaneously within one regression model to predict intelligence $(y = \beta_0 + \beta_1 X_1 + \beta_2 X_2 + \ldots + \beta_{13} X_{13} + \beta_{14} X_{14} + \in$; Supplementary Fig. S14).

**Prediction performance.** The performance of models to predict intelligence was determined as the Pearson correlation between predicted and observed intelligence scores. For robustness, prediction performance was averaged across 100 repetitions with different training-test splits. Statistical significance was examined with non-parametric permutation tests (1000 iterations; $p < 0.05$).

To test for significant differences in prediction performance between all possible pairs of models (9 different models; 36 comparisons), the absolute difference in prediction performance between two models trained on observed intelligence scores ($|\Delta r_{observed}|$) was compared to the absolute difference in prediction performance when those models were trained on permuted scores ($|\Delta r_{permuted}|$). Specifically, $|\Delta r_{permuted}|$ was computed 1000 times, evaluating how often it exceeded $|\Delta r_{observed}|$ (significant $p < 0.05$).

### Replication in lockbox and external samples

To assess robustness and generalizability, all analyses were repeated in a lockbox dataset (30% of the HCP; $N = 232$; 131 female; 212 right-handed; mean age = 28.5 years; age range = 22–36 years) and in two independent samples from the AOMIC[54–56] (PIOP1 and PIOP2; Supplementary Table S24) in which general intelligence was operationalized as sum score of the RAPM Test[59,60]. Image acquisition and preprocessing details also differed from the HCP (Supplementary Information, pp. 48–49). However, participants were excluded following the same criteria, resulting in 126 participants from PIOP1 (70 female; 112 right-handed; mean age = 22.24 years; age range = 18.25–26 years) and 180 participants from PIOP2 (103 female; 160 right-handed; mean age = 21.91 years; age range = 18.25–25.5 years). In all replication samples (lockbox, PIOP1, and PIOP2), SC–FC coupling was operationalized similarly, and all analyses were performed analogously to the main sample.

Finally, for each of the nine prediction models (eight condition-specific Basic NMA Models; one cross-condition Expanded NMA Model), cross-sample model generalization tests were performed: Main sample data were split into five folds, and models were trained on 80% of the data (4/5 folds). This resulted in five prediction models for each of the 100 training-test data splits, which were then used to predict general intelligence in the replication samples. The performance of each cross-sample model generalization test was assessed by averaging across all prediction performance outcomes. Note that due to differences in tasks applied during fMRI assessment, not all cross-sample model generalization tests could be implemented in the AOMIC samples.

### Reporting summary

Further information on research design is available in the Nature Portfolio Reporting Summary linked to this article.

### Data availability

Data from the main and the lockbox samples were obtained from the Human Connectome Project's 1200 Subjects Data Release (S1200)[52,53], which can be accessed through the HCP data platform, ConnectomeDB, under https://www.humanconnectome.org/study/hcp-young-adult. Access requires registration and agreement to data use terms. Replication samples were drawn from the AOMIC[54–56] and are publicly available through OpenNeuro at https://openneuro.org/datasets/ds002785/versions/2.0.0 (PIOP1) and https://openneuro.org/datasets/ds002790/versions/2.0.0 (PIOP2).

### Code availability

All analyses described in this paper were performed using MATLAB (version R2021a) and R (version 4.0.2). Computer code for the analyses, including preprocessing, is available on GitHub: DWI preprocessing: https://github.com/civier/HCP-dMRI-connectome; FMRI preprocessing:

https://github.com/faskowit/app-fmri-2-mat; Computation of latent *g*-factor: https://github.com/jonasAthiele/BrainReconfiguration_Intelligence; Operationalization of SC–FC coupling: https://github.com/brain-networks/local_scfc; Main analysis and replication analysis: https://github.com/johannaleapopp/SC_FC_Coupling_Task_Intelligence. The code specifically developed for the analysis presented in this paper has also been deposited on Zenodo (https://doi.org/10.5281/zenodo.15348080)[109].

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

## Acknowledgements

We thank the Human Connectome Project[52,53], WU-Minn Consortium (Principal Investigators: David Van Essen and Kamil Ugurbil; 1U554MH091657) funded by the 16 NIH Institutes and Centers that support the NIH Blueprint for Neuroscience Research, and by the McDonnell Center for Systems Neuroscience at Washington University, and all contributors to the Amsterdam Open MRI Collection[54–56] for providing open access data. This work was supported by the German Research Foundation [grant number HI 2185—1/3] assigned to K.H., the German National Academic Foundation [funds from the Federal Ministry of Education and Research] assigned to J.L.P., and the Heinrich-Böll Foundation [funds from the Federal Ministry of Education and Research, grant number P145957] assigned to J.A.T. Further, this research was partially supported by Lilly Endowment, Inc., through its funding for the Indiana University Pervasive Technology Institute. Additional support was provided by the Overhead Programme and Equal Opportunities Funding of the Faculty of Human Sciences at the University of Würzburg, the Equal Opportunities Funding of the Institute of Psychology at the University of Würzburg, and the Open Access Publication Fund of the University of Würzburg.

## Author contributions

Conceptualization: J.L.P. and K.H. Data curation: J.F. and J.L.P. Formal analysis: J.L.P. Funding acquisition: J.L.P. and K.H. Methodology: C.S., J.A.T., J.F., J.L.P., K.H., and O.S. Resources: K.H. Supervision: K.H. Visualization: J.L.P. Writing—original draft: J.L.P. and K.H. Writing—review and editing: C.S., J.A.T., J.F., and O.S.

## Funding

## Competing interests

The authors declare no competing interests.

## Ethics approval

All ethical regulations relevant to human research participants were followed. Procedures of the HCP were authorized by the Washington University Institutional Review Board[52], while the ethical committee of the Department of Psychology at the University of Amsterdam[54] approved study protocols of the Amsterdam Open MRI collection. All participants provided informed written consent in accordance with the principles of the Declaration of Helsinki. We acknowledge the significance of inclusivity in scientific research and are dedicated to fostering a diverse research environment, valuing contributions from individuals irrespective of gender, age, race, ethnicity, or socioeconomic background.
