## [Transparent Peer Review file · Communications Biology]

Structural-Functional Brain Network Coupling During Cognitive Demand Reveals Intelligence-Relevant Communication Strategies

Corresponding Author: Dr Kirsten Hilger

Version 0:

Reviewer comments:

Reviewer #1

(Remarks to the Author)

Summary

In this paper, Popp et al. used resting-state and task-based fMRI data to investigate whether the coupling between structural and functional connectivity (i.e., SC-FC coupling as determined via network similarity and communication measures) was predictive of general intelligence scores both at the whole brain and at regional level. They leveraged a robust univariate predictive framework applied to training, testing, and external validation samples, and showed that SC-FC coupling (1) was highest at rest and lowest during emotional processing, and (2) it unfolded along a specific topological gradient with unimodal areas presenting highest and heteromodal areas weakest SC-FC coupling scores. While at the whole brain only SC-FC coupling values in the emotional tasks were predictive of intelligence scores, at the regional level tasks with higher cognitive load were the best predictors of intelligence across individuals. Despite heterogeneity of findings across SC-FC coupling measures and samples (including failed replication of some results), this study brings novel insights into the behavioural relevance of widely adopted large-scale network neuroscience measures.

Overall, I found this paper well written. Study motivation and design were well articulated, with great strengths being the study's preregistration, the adherence to open science practices, the novelty in combining multimodal data prior to statistical analyses, and the openness about the study's limitations. The discussion section was also a pleasure to read. I have however some concerns regarding methodology and some other more general comments about the results. I report them all here below.

Introduction

- 1) There needs to be a better framing of intelligence in the introduction, especially as it pertains to which brain regions are mostly related to intelligence across the various theories presented. This addition is necessary given the region-specific analyses carried out in the present paper.
- 2) It is unclear how the authors picked the SC-FC coupling measures reported in the paper. There needs to be better motivation for their choices, especially given the differences in metrics with their NI 2024 paper.
- 3) While it is an advantage to combine SC and FC prior to statistical analyses, as it stands the motivation for doing so remains pretty weak and unclear. I suggest adding a section in the introduction to strengthen this important point.

Methods:

- 1) There is a lack of clarity around the methods adopted to obtain general intelligence scores. Why did the authors pick the model they picked? What are the loadings of each variable to the general factor? I understand that most literature on intelligence uses bi-factor models, but not all readers may be familiar with this bit of information.
- 2) Adopting a massive univariate predictive framework comes with big weaknesses, as pointed out by the authors. While multiple comparisons corrections and post-hoc tests help mitigate these issues, I wonder what the impact of variable multicollinearity is on the regression results. In this context, other techniques, such as PLS-regression, could be a better analysis choice than B-NMA. As such, it would be important to motivate the use of B-NMA against other predictive frameworks.

Results:

- 1) Figure 4 should be Figure 1, given the complex workflow and the structure of the paper with Methods at the end.

- 2) The nodes should be 358 and not 360, as per the methods section.
- 3) All bar plots are missing error bars.
- 4) Related to my comment in the Methods section, here the reader would benefit from more details about the bi-factor model to interpret task-specific findings. For instance, could one factor/task be loading more onto the g-factor and therefore be contributing more to the brain-behaviour patterns presented?

Discussion:

- 1) I suggest removing any interpretation of results that either are not significant (i.e., motor task), or do not replicate.
- 2) The authors should add content in the limitation section about the reason for failed replication.

Extra Comment:

- 1) If possible, I would suggest to reduce the number of abbreviations in the paper to improve clarity and fluidity.

Reviewer #2

(Remarks to the Author)

In this article, the authors provide an investigation of how the coupling between SC and FC can predict the general intelligence. Overall the results are interesting but there may need more details and additional analyses to strengthen the current manuscript.

1. As it is a general statement, it is necessary to verify the findings on another dataset.
2. It is known that the FC are similar yet slightly in different tasks. Thus it is necessary to tell whether the coupling difference is simply a result from that.
3. The figure of schematic overview should be moved to the front.
4. Covariates including FC and SC should be included into the prediction model. Also, it needs ablation study to tell whether the coupling predicts something different here.
5. It needs additional analysis with gene expression or other potential factors to explain the emergence of the current findings.

Reviewer #3

(Remarks to the Author)

Version 1:

Reviewer comments:

Reviewer #1

(Remarks to the Author)

The authors did a fantastic job on this revision. They addressed all my concerns. I look forward to seeing this manuscript online.

Reviewer #2

(Remarks to the Author)

Thanks to the authors for addressing my comments. Most of my concerns have been addressed except for "Reviewer comment 4: Covariates including FC and SC should be included in the prediction model. Also, it needs an ablation study to tell whether the coupling predicts something different here." When we say coupling, typically we refer to the interaction between variables A and B. Thus it is indeed a nonlinear effect and should not be a linear combination of A and B. This is why we need to include A and B as covariates in the regression model, no matter if the goal is to construct an interpretable model or a SOTA prediction model. We should at least observe a significant existence before we talk about what it is.

Reviewer #3

(Remarks to the Author)

The authors have done an admirable job addressing the reviewer comments. I am satisfied by not only their responses to my comments, but the by the way they have addressed other reviewers suggestions. I do not have any additional comments.

Below please find our detailed responses to the Reviewers' comments. We are extremely grateful for the overall very positive and encouraging feedback we have received from the Reviewers and the Editor and have attempted to accommodate all issues thoroughly.

Reviewer #1:

In this paper, Popp et al. used resting-state and task-based fMRI data to investigate whether the coupling between structural and functional connectivity (i.e., SC-FC coupling as determined via network similarity and communication measures) was predictive of general intelligence scores both at the whole brain and at regional level. They leveraged a robust univariate predictive framework applied to training, testing, and external validation samples, and showed that SC-FC coupling (1) was highest at rest and lowest during emotional processing, and (2) it unfolded along a specific topological gradient with unimodal areas presenting highest and heteromodal areas weakest SC-FC coupling scores. While at the whole brain only SC-FC coupling values in the emotional tasks were predictive of intelligence scores, at the regional level tasks with higher cognitive load were the best predictors of intelligence across individuals. Despite heterogeneity of findings across SC-FC coupling measures and samples (including failed replication of some results), this study brings novel insights into the behavioural relevance of widely adopted large-scale network neuroscience measures.

Overall, I found this paper well written. Study motivation and design were well articulated, with great strengths being the study's preregistration, the adherence to open science practices, the novelty in combining multimodal data prior to statistical analyses, and the openness about the study's limitations. The discussion section was also a pleasure to read. I have however some concerns regarding methodology and some other more general comments about the results. I report them all here below.

We thank the Reviewer for their overall very positive evaluation and appreciate their recognition of our adherence to open science principles, the novelty in combining multimodal data prior to statistical analyses, and our discussion of limitations. We also value the Reviewer's specific and highly constructive comments and suggestions, which we address in detail below.

Introduction

Reviewer comment 1:

There needs to be a better framing of intelligence in the introduction, especially as it pertains to which brain regions are mostly related to intelligence across the various theories presented. This addition is necessary given the region-specific analyses carried out in the present paper.

Author's response 1:

We completely agree with the Reviewer that important details about intelligence-related brain regions were missing in the former version of our Introduction section. They accurately point out that, especially in the context of our prediction framework that uses region-specific SC-FC coupling data, this information would strengthen the rationale for our study. In response, we have expanded the Introduction section:

Line 98 – 109 (Introduction):

“Individual differences in general intelligence have been associated with brain structure^{7,8} and brain function^{9,10} (for review see Hilger et al.¹¹). Dominant neuro-cognitive models of

intelligence, such as the Parieto-Frontal Integration Theory¹² and the Multiple Demand System¹³, suggest a key role of frontal and parietal regions along with additional involvement of brain regions in temporal and occipital cortices. A more recent meta-analysis confirmed the importance of these regions, while also proposing critical relevance of the insular cortex, posterior cingulate cortex, and subcortical structures¹⁴. Further, the Neural Efficiency Hypothesis initially suggested that more intelligent individuals exhibit lower, thus more efficient, activation of predominantly frontal brain regions during cognitive tasks¹⁵. However, subsequent reevaluation identified moderating variables such as sex, task type, and task complexity¹⁶. Finally, most recent proposals subsume evolving evidence from network neuroscience studies and conclude that distinct attributes of structural and functional brain networks are critical for understanding individual differences in general intelligence^{17,18}.”

Reviewer comment 2:

It is unclear how the authors picked the SC-FC coupling measures reported in the paper. There needs to be better motivation for their choices, especially given the differences in metrics with their NI 2024 paper.

Author’s response 2:

We thank the Reviewer for this entirely valid notion. It is indeed absolutely true that, in the previous version of the manuscript, we have not sufficiently explained a) why we reduced the number of SC-FC coupling measures compared to our NeuroImage paper (Popp et al., 2024) and b) how we selected the similarity measure (cosine similarity) and the three communication measures (path length, communicability and search information).

In our 2024 paper, we applied eight coupling measures to examine the relationship between resting-state SC-FC coupling and general cognitive ability, i.e., coupling was only assessed in one single condition (resting state). In contrast, in our current study, we investigate SC-FC coupling across eight conditions, introducing additional comparisons and overall complexity. To minimize the danger of overfitting (the more variables involved in a model, the higher the likelihood of fitting sample-specific aspects or even noise) and to facilitate the straightforward interpretation of findings, we believed that selecting a theoretically justified subset of coupling measures would be beneficial.

In general, the measures used to operationalize SC-FC coupling in the 2024 paper can be grouped into four categories based on the proposed signaling mechanism.

- 1) Routing (path length)*
- 2) Path accessibility (path transitivity, search information)*
- 3) Diffusion (communicability, mean first passage time, flow graphs)*
- 4) Similarity (cosine distance, matching index)*

As measures within the same category may capture overlapping information, in the current study, we chose one representative measure from each category. This selection of one measure per category was performed in a data-driven manner, based on which measure - within a given category - explained the highest variance in functional connectivity (see Figure R1 for the resting-state condition as example). This selection procedure was preregistered.

Figure R1: Data-driven selection of communication measures. Specifically, this Figure illustrates the measure that was able to explain the highest R^2 in the regional connectivity profile of the FC matrix during the resting-state condition across all individuals in the main sample ($N = 532$) the most often i.e., was chosen as ‘best measure’ with the highest frequency. Corresponding brain maps for the seven task conditions appeared very similar. Thus, path length (PL), communicability (G), cosine similarity (CoS) and search information (SI) were chosen as representative measure for their category.

We agree with the Reviewer that this information is valuable for the reader and have, therefore, included it in the Introduction and the Methods sections:

Line 156 – 159 (Introduction):

“To address these gaps, we used data from 764 participants of the Human Connectome Project (HCP⁵²) to investigate SC-FC coupling operationalized with one similarity and three communication measures, capturing the range of signaling strategies from routing to diffusion^{39,45}, during different conditions.”

Line 704 – 712 (Methods):

“Based on each individual’s SC matrix, one similarity measure (cosine similarity, CoS) and three communication measures (path length, PL; communicability, G; search information, SI) were computed^{32,40} (Table 2; details in Supplementary Information pp. 46-48). These measures were selected from multiple alternatives³², as they are representative of four signaling categories (routing, path accessibility, diffusion, and similarity^{39,45}). Brain region-specific coupling values were then obtained by correlating regional connectivity profiles (matrix

columns) of each similarity or communication matrix with the respective regional connectivity profile of a condition-specific FC matrix (4*8 comparisons). This resulted in 358 region-specific coupling values (r_c) for each comparison (for each fMRI condition and each measure; Fig. 1a-d) per participant.”

Reviewer comment 3:

While it is an advantage to combine SC and FC prior to statistical analyses, as it stands the motivation for doing so remains pretty weak and unclear. I suggest adding a section in the introduction to strengthen this important point.

Author’s response 3:

We thank the Reviewer for their valuable feedback and appreciate the suggestion to further highlight the strengths of combining SC and FC prior to statistical analyses. As the Reviewer noted correctly, this pre-analysis integration is a strength of our work, as any reduction in the number of variables within a statistical model lowers the danger of overfitting and improves computational efficiency. Most importantly though, by assessing the alignment of structural and functional brain networks, we create a biologically meaningful measure that allows for insights concerning the interplay between structural pathways and functional communication. We have extended the respective paragraph in the Introduction section accordingly:

Line 111 – 125 (Introduction):

*“While structural brain networks approximate physical connections between brain regions (axons and dendrites), functional brain networks are constructed based on statistical relationships between time courses of neural activation¹⁹. Characteristics of both structural connectivity (SC) and functional connectivity (FC) have been related to intelligence²⁰⁻²⁴ and even allow for significant prediction of individual intelligence scores in cross-validated predictive modeling approaches^{25,26} (for review see Hilger et al.¹¹; Hilger & Sporns¹⁸). However, although the best predictions of intelligence were achieved by system-wide approaches integrating multiple neuroimaging modalities²⁷⁻³⁰, the potential to predict intelligence from variables unifying different modalities before **entering a statistical model, (i.e., feature-level multi-modality) remains widely unknown. Here, we assess the alignment between structural and functional brain networks, referred to as SC-FC coupling. Combining SC and FC prior to statistical analyses offers key advantages, as any reduction in the number of variables within a statistical model lowers the danger of overfitting and improves computational efficiency³¹. Most importantly, by assessing the alignment of structural and functional brain networks, we create a biologically meaningful measure that allows for insights concerning the interplay between structural pathways and functional communication³².”***

Methods

Reviewer comment 4:

There is a lack of clarity around the methods adopted to obtain general intelligence scores. Why did the authors pick the model they picked? What are the loadings of each variable to the general factor? I understand that most literature on intelligence uses bi-factor models, but not all readers may be familiar with this bit of information.

Author’s response 4:

We thank the Reviewer for bringing this lack of clarity regarding the methods used to obtain general intelligence scores to our attention and would like to provide additional details.

General intelligence (g) is inherently defined as a latent factor that captures the common variance across different cognitive tasks (Spearman, 1904). Thus, we assumed that modelling g as latent factor derived from the performance scores of all available cognitive tasks may provide a more valid estimate than a single test score or sum of scores.

*We applied a bi-factor model for several key reasons: On the one hand, cognitive test scores load onto a **general factor (g-factor)**, which represents the common variance across all cognitive abilities. On the other hand, test scores also load onto **domain factors** which capture the variance unique to each cognitive domain (e.g., verbal, spatial, memory). In contrast to traditional hierarchical models (i.e., higher-order models), where the domain factors mediate the relationship between g-factor and subtest scores (pyramidal structure), in bi-factor models the domain factors are modeled as independent of the g-factor. This independence not only allows for the direct assessment of how each cognitive test score loads onto the g-factor, it also enables a clearer separation of general intelligence from specific cognitive abilities reflected in the domain factors (Cucina & Byle, 2017).*

Importantly, bi-factor models have been shown to provide a better fit to similar cognitive test data than alternative models and are, thus, suggested to more accurately capture the structure of intelligence compared to higher-order models (Cucina & Byle, 2017; Kan et al., 2024).

Finally, as the Reviewer already correctly pointed out, the use of bi-factor models is standard in intelligence research (Dubois et al., 2018). Following this standard facilitates comparisons between our work and prior neuroimaging studies, some of which even use the same dataset (HCP; e.g., Dubois et al., 2018; Thiele et al., 2022, 2024).

In response to the Reviewer's comment, we have clarified our rationale behind using bi-factor modeling in the Methods section. Also, we have included additional information in the Results Section and a Supplementary Figure that specifies the loadings of each cognitive test score onto the g-factor.

Line 661 – 664 (Methods):

“General intelligence was computed as latent g-factor from 12 cognitive measures^{32,62} (Supplementary Table S1) with bi-factor analysis^{25,54} using data from 1186 participants of the HCP⁵². This approach is standard in intelligence research^{25,62,89} and has been shown to offer a better fit to diverse cognitive test scores than alternative models^{94,95}.”

Line 171 – 176 (Results):

“General intelligence was operationalized as latent g-factor from 12 cognitive measures (Supplementary Table S1) using bi-factor analysis^{25,54}. Details about the model, including factor loadings of each cognitive measure onto the g-factor, are reported in Supplementary Fig. S1. The data of the HCP⁵² was divided into a main sample (N = 532) and a smaller subsample for later lockbox replication (N = 232). Scores in the main sample ranged between

-2.38 and 2.40 ($M = .25$; $SD = .83$) and their frequency distribution is visualized in Supplementary Fig. S2a, suggesting normality.”

Supplementary Information

“Supplementary Fig. S1 I Estimation of general intelligence from 12 cognitive scores using bi-factor analysis. General intelligence was computed as latent g-factor from 12 cognitive scores (Supplementary Table S1) with bi-factor analysis^{1,2}. According to the model fit criteria established by Hu and Bentler³, the 4-bi-factor model fits the data well (Comparative Fit Index: CFI = 0.979, Root Mean Square Error of Approximation: RMSEA = 0.0395, and Standardized Root Mean Square Residual: SRMR = 0.0213)⁴.”

Reviewer comment 5:

Adopting a massive univariate predictive framework comes with big weaknesses, as pointed out by the authors. While multiple comparisons corrections and post-hoc tests help mitigate these issues, I wonder what the impact of variable multicollinearity is on the regression results. In this context, other techniques, such as PLS-regression, could be a better analysis choice than B-NMA. As such, it would be important to motivate the use of B-NMA against other predictive frameworks.

Author’s response 5:

We fully understand these concerns and are happy to elaborate on our choice more in detail. As the Reviewer correctly assumes, some multicollinearity exists between the variables that are incorporated into the B-NMA prediction framework, i.e., between the average coupling value extracted based on the positive NMA and the average coupling value extracted based on the negative NMA. This arises from our feature generation process, where individual region-specific coupling values derived from the positive or negative NMA mask only differ in the assigned coupling measure per brain region, but both capture the amount of coupling.

In classical regression models, multicollinearity might be problematic, as it can lead to unstable regression coefficients. However, this is not problematic in our case, as we are not interpreting single regression coefficients but rather evaluate overall prediction performance. If multicollinearity was an issue, it would manifest in poor model generalization, reflected in lower correlation coefficients between the predicted and observed intelligence scores. As long as the predictive performance remains stable, and the model generalizes well – which is the case on our work - multicollinearity should not be a major concern.

Given the complexity of the SC-FC coupling operationalization, we opted for the B-NMA prediction framework due to its simplicity and interpretability. More sophisticated approaches, such as PLS regression, would introduce latent variables and additional transformations that may obscure the relationship between SC-FC coupling and intelligence, making interpretation more challenging. Next, our framework is computationally efficient, enabling thorough cross-validation and extensive replication. Importantly, the B-NMA was carefully developed and validated in our prior work (Popp et al., 2024), demonstrating reliability in predicting general cognitive ability from intrinsic SC-FC coupling. By employing this framework, we ensure consistency with our previous research, while extending the investigation to task conditions.

However, we agree with the Reviewer that investigating the relationship between SC-FC coupling and intelligence using alternative prediction models – both of linear and non-linear nature - represents an interesting subject for future study and included a respective notion in the Discussion section:

Line 600 – 623 (Discussion):

*“There are limitations to this study. First, our samples were restricted to young adults, posing the question of result-generalizability to a broader age range. Second, neuroimaging data are inherently noisy⁸⁵, and both FC and SC have limited reliability^{86,87}. Although we applied state-of-the-art methods addressing these issues, remaining influences of confounds cannot be ruled out. Third, our analyses focused on one similarity and three communication measures *as well as the application of two specific prediction frameworks (B-NMA, E-NMA)*, while additional measures^{39,40,63} and alternative prediction frameworks (i.e., PLS-regression^{27,88}; neural networks⁸⁹) are available. Although the here developed framework is computationally very efficient and allows for intuitive interpretations, the systematic evaluation of our framework against alternative methods may offer further insights into the potential of our approach. Especially approaches combining different communication strategies⁶³ and methods that account for potential non-linear relations may offer promising starting points for future research on the relationship between intelligence and SC-FC coupling. Fourth, task variety and difficulty are limited. As it has been shown that relations between intelligence and neural activation can differ depending on task content and difficulty^{90,91}, expanding both may provide complementary insights. Fifth, it is important to acknowledge that although reaching significance in the main sample, not all of our results replicated in the lockbox and the independent samples. Possible reasons include smaller sample sizes (lockbox and AOMIC samples) as well as differences in a) scanner sites, b) image acquisition parameters, c) task*

conditions, and d) data quality in the AOMIC samples. Finally, forthcoming studies could profit from a) investigating the relationship between general intelligence and individual SC-FC coupling changes over longer time periods where SC underlies significant changes, b) disentangling the individual contributions of SC and FC to the predictive power of SC-FC coupling measures, and c) incorporating naturalistic in-scanner stimuli (e.g., movie paradigms), as these may posit a valuable compromise between overly constrained tasks lacking ecological validity and unrestricted resting state, thus potentially amplifying behaviorally relevant individual differences^{47,48}.”

Results

Reviewer comment 6:

Figure 4 should be Figure 1, given the complex workflow and the structure of the paper with Methods at the end.

Author’s response 6:

We thank the Reviewer for this comment and fully agree that changing the Figure order will help the reader to better comprehend the complex workflow. We have made these changes accordingly, and referred to the respective Figure at the beginning of the Results section:

Line 183 – 187 (Results):

*“Individual structural brain networks were transformed into one similarity and three communication measures (Table 2) before computing region-specific SC-FC coupling values, defined as the Pearson correlation between regional connectivity profiles of the similarity or communication networks and the individual condition-specific functional networks. **Please note that a schematic overview of the complete study procedure is presented in Fig. 1.**”*

Reviewer comment 7:

The nodes should be 358 and not 360, as per the methods section.

Author’s response 7:

We thank the Reviewer for bringing this to our attention and apologize for this mistake. We have corrected this information in the Results section:

Line 180 – 182 (Results):

*“To operationalize the structural-functional brain network coupling (SC-FC coupling), structural brain networks were constructed from diffusion weighted imaging, while functional brain networks were derived from eight fMRI conditions (Table 1) using a multimodal parcellation scheme (**358 nodes**⁵⁵).”*

Reviewer comment 8:

All bar plots are missing error bars.

Author’s response 8:

We apologize that we had missed this important detail in our Figures. We have updated all bar plots accordingly, i.e., to include error bars, and now also illustrate all data points to allow for insights into the data distribution. We also adjusted color transparency of all individual bars to allow for better visibility of the individual datapoints. These changes were implemented in all bar plots, but Supplementary Fig. S3, as this Figure is simply a visual representation of percentages and there is no underlying data distribution that can be shown. Below, you can find the updated Figures.

Line 338-366 (Results):

“Fig. 3 | Performance to predict intelligence from SC-FC coupling during different fMRI conditions (main sample, N = 532). Predictions were cross-validated (5-fold) and are based on brain region-specific coupling from eight separate fMRI conditions (Basic NMA Model) as well as from task-combined coupling (Expanded NMA Model). *Bar graphs depict mean prediction values across 100 repetitions with different training-test splits and error bars indicate the standard error of the mean.* Significance of prediction performance was assessed with non-parametric permutation tests, while significant differences in prediction performance were assessed with model difference tests (see Methods). P-values indicating significant predictions are marked with an asterisk a) below bar graphs for single models and b) above bar graphs for between-model differences (* = $p < .05$). RES = resting state; EMO = emotion processing task; GAM = gambling task; LAN = language task; MOT = motor task; REL = relational processing task; SOC = social cognition task; WM = working memory task; ALL = all task fMRI conditions.”

Supplementary Information:

“Supplementary Fig. S11 I Performance to predict intelligence from SC-FC coupling during different fMRI conditions (Basic NMA Model) and a combination of all task fMRI conditions (Expanded NMA Model) in the cross-sample model generalization test in the lockbox sample (HCP532 → HCP232). The significance of a prediction performance was assessed with a non-parametric permutation test. Bar graphs depict mean prediction values across 100 repetitions with different splits of training data in the main sample and error bars indicate the standard error of the mean. Significant differences in model performance were assessed with model difference tests (see Methods). P-values indicating significant associations are marked with an asterisk a) below bar graphs for single prediction performance and b) above bar graphs for differences in prediction performance (* = $p < .05$). RES = resting state; EMO = emotion processing task; GAM = gambling task; LAN = language task; MOT = motor task; REL = relational processing task; SOC = social cognition task; WM = working memory task; ALL = all task fMRI conditions.”

Reviewer comment 9:

Related to my comment in the Methods section, here the reader would benefit from more details about the bi-factor model to interpret task-specific findings. For instance, could one factor/task be loading more onto the g-factor and therefore be contributing more to the brain-behaviour patterns presented?

Author’s response 9:

We sincerely appreciate the Reviewer’s valuable suggestion. As noted in our response to their fourth comment (Methods section), we have already addressed this point by incorporating

additional details in the Methods section, the Results section and the Supplementary Information. Our revisions clarify how individual tasks load onto the g-factor, and we hope these additions adequately address the Reviewer's concern.

Discussion

Reviewer comment 10:

I suggest removing any interpretation of results that either are not significant (i.e., motor task), or do not replicate.

Author's response 10:

*We fully agree with the Reviewer that only significant and replicated results should be interpreted in the Discussion section. However, since replication analyses in the AOMIC samples were not always possible due to differences in available fMRI conditions (tasks), we decided to interpret all findings that were a) significant **and** b) replicated at least in the lockbox sample (if not internally cross-validated). In the single instance where we discussed a result that was significant in the main sample, but the lockbox replication did not reach significance (brain-average SC-FC coupling during the emotion processing task is significantly associated with g), we only offer a very cautious interpretation and explicitly highlight the lack of successful replication.*

With this approach, we aimed to provide the reader with as many (hypothesis-generating) insights as possible while simultaneously ensuring full transparency. Especially regarding the predictive modeling framework, we considered this as justified, given that it was internally cross-validated, which reduces the risk of reporting only sample-specific effects - more so than in purely correlative studies.

We are unsure, why the Reviewer mentions findings related to the motor task, since we did not mention respective findings in the Discussion section at all. However, we noted an unclarity in the second paragraph of the Discussion section, where we actually interpreted condition-specific differences in brain-average SC-FC coupling, which might have led to this confusion. In the previous version of the Manuscript, we noted that "Given that these findings could not be replicated in the lockbox nor in the independent AOMIC samples, further research is required to draw stronger conclusions (Previous Manuscript Version: Line 363 – 365)". However, these findings actually were replicated in the lockbox sample (Supplementary Fig. S5; Supplementary Table S6). We deeply apologize for any resulting confusion and have adjusted the statement accordingly to: "Given that these findings could not be replicated in the independent AOMIC samples, further research is required to draw stronger conclusions (Line 516 – 518)".

In response to the Reviewer's comment, we carefully reviewed the entire Discussion section to more explicitly clarify when the replication of interpreted results was possible and when it was not. Again, we apologize for the misunderstanding and hope to have resolved this issue.

Line 539 – 547 (Discussion):

“On a brain-average level, functional interactions operate more closely along structural pathways in more intelligent individuals only during the emotion processing task, which is suggested to involve relatively low cognitive demand^{68,69}. This was consistently observed across all coupling measures and may indicate an efficient use of the underlying structure without the need for more fine-drawn regional adaptations in the face of low demand (in line with^{15,16}). As this would help saving cognitive resources necessary for handling potential future higher demands, such a coupling organization could be interpreted as being adaptive, emphasizing that the ability for efficient adaptation presents a hallmark of human intelligence⁷⁰. *Since replication of this finding was neither possible in the lockbox nor the independent AOMIC samples, further research is needed to reach more definitive conclusions.*”

Line 566 – 576 (Discussion):

“Extending previous reports that demonstrate enhanced prediction of behavioral phenotypes when integrating functional connectivity across multiple tasks^{27,75,76}, combining individual coupling information across tasks descriptively yielded the best prediction performance *in the internal cross-validation in the main sample*. Note that this must not merely be an effect of including more factors in the statistical model, as more factors increase the likelihood for overfitting, which we prevent with cross-validation, and would therefore reduce prediction accuracy⁷⁷. Regarding general intelligence, this observation is particularly interesting, as the g-factor theory proposes the existence of a latent intelligence factor that captures the common variance of an individual’s performance across tasks². Thus, neural processes that might reflect g at the brain level (‘neuro-g’^{78,79}) should also be involved in all tasks⁸⁰. Our study *provides initial evidence that task-combined brain region-specific structural-functional brain network coupling could be a potential candidate for such a proposed ‘neuro-g’.*”

Reviewer comment 11:

The authors should add content in the limitation section about the reason for failed replication.

Author’s response 11:

We appreciate the Reviewer’s suggestions and fully agree that adding potential reasons for the only partially successful replication, such as smaller sample sizes (lockbox sample; AOMIC samples), and differences in a) scanner site, b) image acquisition parameters, c) task conditions, and d) data quality in the AOMIC samples as well as a notion of what this actually means for any interpretations of our findings, to the limitations section represents a valuable addition to the manuscript. We have revised the paragraph accordingly:

Line 600 – 623 (Discussion):

“There are limitations to this study. First, our samples were restricted to young adults, posing the question of result-generalizability to a broader age range. Second, neuroimaging data are inherently noisy⁸⁵, and both FC and SC have limited reliability^{86,87}. Although we applied state-of-the-art methods addressing these issues, remaining influences of confounds cannot be ruled out. Third, our analyses focused on one similarity and three communication measures *as well as the application of two specific prediction frameworks (B-NMA, E-NMA)*, while additional

measures^{39,40,63} and alternative prediction frameworks (i.e., PLS-regression^{27,88}; neural networks⁸⁹) are available. Although the here developed framework is computationally very efficient and allows for intuitive interpretations, the systematic evaluation of our framework against alternative methods may offer further insights into the potential of our approach. Especially approaches combining different communication strategies⁶³ and methods that account for potential non-linear relations may offer promising starting points for future research on the relationship between intelligence and SC-FC coupling. Fourth, task variety and difficulty are limited. As it has been shown that relations between intelligence and neural activation can differ depending on task content and difficulty^{90,91}, expanding both may provide complementary insights. Fifth, it is important to acknowledge that although reaching significance in the main sample, not all of our results replicated in the lockbox and the independent samples. Possible reasons include smaller sample sizes (lockbox and AOMIC samples) as well as differences in a) scanner sites, b) image acquisition parameters, c) task conditions, and d) data quality in the AOMIC samples. Finally, forthcoming studies could profit from a) investigating the relationship between general intelligence and individual SC-FC coupling changes over longer time periods where SC underlies significant changes, b) disentangling the individual contributions of SC and FC to the predictive power of SC-FC coupling measures, and c) incorporating naturalistic in-scanner stimuli (e.g., movie paradigms), as these may posit a valuable compromise between overly constrained tasks lacking ecological validity and unrestricted resting state, thus potentially amplifying behaviorally relevant individual differences^{47,48}.”

Extra Comment

Reviewer comment 12:

If possible, I would suggest to reduce the number of abbreviations in the paper to improve clarity and fluidity.

Author's response 12:

We acknowledge the Reviewers concern and have minimized the use of abbreviations throughout our manuscript:

Particularly, we refrained from using abbreviations for blood oxygen level dependent (BOLD) signal, framewise displacement (FD), Parieto-Frontal Integration Theory (P-FIT), time to echo (TE) and time to repetition (TR) in the revised manuscript. Additionally, we also avoided the utilization of abbreviations for the coupling measures and the task conditions. However, we retained these abbreviations in the Tables and Figures to maintain clarity and organization.

Additionally, we removed all abbreviations from the list in the manuscript that appear exclusively in the Supplementary Information and compiled them into a separate list.

Reviewer #2:

In this article, the authors provide an investigation of how the coupling between SC and FC can predict the general intelligence. Overall the results are interesting but there may need more details and additional analyses to strengthen the current manuscript.

We thank the Reviewer for their overall positive evaluation of our work and appreciate their feedback and constructive comments.

Reviewer comment 1:

As it is a general statement, it is necessary to verify the findings on another dataset.

Author's response 1:

*We sincerely thank the Reviewer for this valuable feedback and apologize if we did not make it sufficiently clear in the previous version of our manuscript that we **had already** verified our findings on even **more than one other dataset!** More in detail, we have conducted extensive replication analyses by replicating our findings in a) a lockbox sample ($N = 232$, i.e., a part of the complete HCP sample which was set aside before any analyses were conducted) and b) two completely independent samples from the Amsterdam Open MRI collection (PIOP1: $N = 126$; PIOP2: $N = 180$). These verifications provide strong evidence for the robustness and generalizability of our findings.*

We thank the Reviewer for bringing to our attention to the issue that it was not sufficiently described in the previous version of the manuscript that we already had performed extensive validation analyses. This is a very important point, and we have revised the relevant sections throughout the text to ensure clarity and resolve any potential uncertainties.

Line 156 – 167 (Introduction):

*“To address these gaps, we used data from **764 participants** of the Human Connectome Project (HCP⁵²) to investigate SC-FC coupling operationalized with one similarity and three communication measures, **capturing the range of signaling strategies from routing to diffusion**^{39,45}, during different conditions. First, task-induced SC-FC coupling was descriptively characterized on a brain-average and brain region-specific level and then related to intrinsic (resting state) coupling. Next, the relationship between general intelligence and brain-average SC-FC coupling was examined with a correlative approach and region-specific SC-FC coupling was tested for its potential to predict individual intelligence scores within a cross-validated predictive modeling framework. Finally, since general intelligence is defined as latent capability influencing performance on various cognitive tasks, it was explored whether combining coupling information from multiple tasks can enhance prediction performance. **All analyses were repeated in a lockbox sample from the HCP and two completely independent samples from the Amsterdam Open MRI collection (AOMIC⁵³).**”*

Line 400 – 403 (Results):

*“To evaluate the robustness **and generalizability** of our findings, we **performed extensive validation analyses**. Specifically, we conducted the same analyses (**replication**) in a lockbox sample from the HCP ($N = 232$)⁵² and in two completely independent samples from the AOMIC (AOMIC PIOP1, $N = 126$; AOMIC PIOP2, $N = 180$)⁵³.”*

Line 495 – 497 (Discussion):

“We used fMRI data acquired during different conditions ($N_{main} = 532$, *replication*: $N_{lockbox} = 232$; $N_{PIOP1} = 126$; $N_{PIOP2} = 180$) and operationalized SC-FC coupling with one similarity and three communication measures.”

Line 798 – 802 (Methods):

“To assess robustness and generalizability, all analyses were *repeated* in a lockbox dataset (30% of the HCP; $N = 232$; 131 female; 212 right-handed; mean age = 28.5 years; age range = 22-36 years) and in two independent samples from the AOMIC⁵³ (PIOP1 & PIOP2; Supplementary Table S24) in which general intelligence was operationalized as sum score of the Raven’s Advanced Progressive Matrices Test (RAPM)^{56,57}.”

Reviewer comment 2:

It is known that the FC are similar yet slightly in different tasks. Thus it is necessary to tell whether the coupling difference is simply a result from that.

Author’s response 2:

*We thank the Reviewer for this comment and would like to clarify that their assumption is, of course, entirely correct. Since structural connectivity serves as a proxy for actual physical connections in the brain (i.e., axon fibers; white matter tracts), it remains relatively stable over short time periods, such as the duration of an MRI recording (Osmanlioglu et al., 2020). Any differences in SC-FC coupling between conditions may thus solely arise from variations in functional connectivity, which also fluctuates over shorter time periods (Gonzalez-Castillo & Bandettini, 2018). This is clarified in Figure 1: Specifically, when computing individual region-specific SC-FC coupling values for each condition, similarity and communication measures are computed from a **single** SC matrix per participant and then compared to each of the **multiple** condition-specific FC matrices.*

As a closer examination of this issue may offer valuable directions for future research, such as investigating the relationship between individual variations in general intelligence and individual SC-FC coupling changes over longer time periods - where structural connections may significantly change. We now refer to this point in the Discussion section and thank the Reviewer again for their valuable thoughts:

Line 600 – 623 (Discussion):

“There are limitations to this study. First, our samples were restricted to young adults, posing the question of result-generalizability to a broader age range. Second, neuroimaging data are inherently noisy⁸⁵, and both FC and SC have limited reliability^{86,87}. Although we applied state-of-the-art methods addressing these issues, remaining influences of confounds cannot be ruled out. Third, our analyses focused on one similarity and three communication measures *as well as the application of two specific prediction frameworks (B-NMA, E-NMA)*, while additional measures^{39,40,63} and alternative prediction frameworks (i.e., PLS-regression^{27,88}; neural networks⁸⁹) are available. Although the here developed framework is computationally very efficient and allows for intuitive interpretations, the systematic evaluation of our framework against alternative methods may offer further insights into the potential of our approach. Especially approaches combining different communication strategies⁶³ and methods that

account for potential non-linear relations may offer promising starting points for future research on the relationship between intelligence and SC-FC coupling. Fourth, task variety and difficulty are limited. As it has been shown that relations between intelligence and neural activation can differ depending on task content and difficulty^{90,91}, expanding both may provide complementary insights. Fifth, it is important to acknowledge that although reaching significance in the main sample, not all of our results replicated in the lockbox and the independent samples. Possible reasons include smaller sample sizes (lockbox and AOMIC samples) as well as differences in a) scanner sites, b) image acquisition parameters, c) task conditions, and d) data quality in the AOMIC samples. Finally, forthcoming studies could profit from a) investigating the relationship between general intelligence and individual SC-FC coupling changes over longer time periods where SC underlies significant changes, b) disentangling the individual contributions of SC and FC to the predictive power of SC-FC coupling measures, and c) incorporating naturalistic in-scanner stimuli (e.g., movie paradigms), as these may posit a valuable compromise between overly constrained tasks lacking ecological validity and unrestricted resting state, thus potentially amplifying behaviorally relevant individual differences^{47,48}.”

Reviewer comment 3:

The figure of schematic overview should be moved to the front.

Author’s response 3:

We appreciate the Reviewer's comment and completely agree that adjusting the figure order will be beneficial for the reader's understanding of our complex workflow. We have revised the figure order and now already refer now to the schematic overview in the Results section.

Line 183 – 187 (Results):

“Individual structural brain networks were transformed into one similarity and three communication measures (Table 2) before computing region-specific SC-FC coupling values, defined as the Pearson correlation between regional connectivity profiles of the similarity or communication networks and the individual condition-specific functional networks. Please note that a schematic overview of the complete study procedure is presented in Fig. 1.”

Reviewer comment 4:

Covariates including FC and SC should be included into the prediction model. Also, it needs ablation study to tell whether the coupling predicts something different here.

Author’s response 4:

We thank the Reviewer for this comment and appreciate the opportunity to clarify our methodology. While we understand their perspective, we respectfully disagree with their first statement. Our measure of interest, SC-FC coupling, is derived from both SC and FC. It reflects their alignment and controlling for FC and SC would actually destroy the concept we are interested in. In our opinion, this would obscure rather than clarify the role of SC-FC coupling in intelligence.

Rather than including SC and FC as covariates, it would, in our opinion, be more meaningful to evaluate the ability of SC and FC individually to predict intelligence and then compare their prediction performance to that of SC-FC coupling. We believe that this aligns with the second part of the Reviewer's comment. While we agree that this would be interesting, it is important to note that such analyses have already been extensively conducted in prior research, including studies using the same sample (intelligence prediction from SC: Kopetzky et al., 2024; Zhang et al., 2019; intelligence prediction from FC: Dubois et al., 2018; Thiele et al., 2024). We would like to highlight that our goal with this research was not to outperform these studies with respect to prediction performance, but rather to develop a biologically interpretable feature and explore its relationship with intelligence.

In response to the Reviewer's comment, we have incorporated this aspect into our discussion of the study's limitations:

Line 617 – 623 (Results):

“Finally, forthcoming studies could profit from a) investigating the relationship between general intelligence and individual SC-FC coupling changes over longer time periods where SC underlies significant changes, b) disentangling the individual contributions of SC and FC to the predictive power of SC-FC coupling measures, and c) incorporating naturalistic in-scanner stimuli (e.g., movie paradigms), as these may posit a valuable compromise between overly constrained tasks lacking ecological validity and unrestricted resting state, thus potentially amplifying behaviorally relevant individual differences^{47,48}.”

Reviewer comment 5:

It needs additional analysis with gene expression or other potential factors to explain the emergence of the current findings.

Author's response 5:

This is absolutely true! Future studies also incorporating genetic data could offer additional insights into the emergence of the here reported relationship between SC-FC coupling and intelligence.

It is well-established that there exists a substantial genetic influence on individual differences in intelligence (Deary et al., 2009). However, genes do not directly shape human traits and behavior, rather they act through the neurobiology of a person's brain, i.e., brain structure and/or brain function to contribute to individual differences in trait and behavior (see i.e., Haier, 2023). The identification of robust neurobiological correlates of intelligences that may function as those intermediate phenotypes, such as SC-FC coupling, presents a crucial first step in disentangling the complex relationship between genetic makeup and individual differences in intelligence. Once these foundational neurobiological links are established through further studies, genetic analyses may provide the missing piece of the puzzle to understanding how individual differences in brain neurobiology underlying individual differences in intelligence evolve.

Although these additional genetic analyses are far beyond the scope of our current paper, we actually plan to address this topic in future research. We have already submitted a research proposal to apply for funding to conduct these analyses. We hope that the proposal will be accepted, allowing us to present follow-up studies that combine SC-FC coupling and genetic data in the near future.

Reviewer #3:

The manuscript examines the relationship between structural-functional brain network coupling and general intelligence. The manuscript uses data from the HCP, including independent replication, as well as replication data from the AOMIC data set. The authors find that structural-functional coupling was highest during resting state and lowest during the emotion task. They replicate previous studies showing that coupling is higher in unimodal brain regions relative to multimodal areas. They found at a whole-brain level that coupling is related to intelligence only during the emotion task. Analyses focused on the regional level found that coupling was significantly related to general intelligence for all the tasks except for the motor task. This is an impressive manuscript that tackles a novel question and will be of interest to a wide audience. The analyses are appropriate, and have sufficient explanation that they can be reproduced by other researchers. I have a couple relatively minor suggestions that could potentially improve the manuscript provided below:

We sincerely appreciate the Reviewer's positive assessment of our work and the genuinely constructive comments.

Reviewer comment 1:

The manuscript calculates FC during several tasks. The method includes basis-set task regressors to remove average task activation prior to calculating the correlation. I think this is a reasonable approach. The authors cite Cole et al. (2019), but that paper found the basis set approach was less effective than an FIR approach. It would be useful if the authors justified their choice to use the basis set approach in the current context.

Author's response 1:

We thank the Reviewer for this comment and welcome the opportunity to justify our choice of using the basis set approach to remove task activation.

First, we would like to highlight that in the Discussion section of Cole et al. (2019), the authors explicitly state that both approaches (FIR and basis set GLM) work quite well for the given purpose: "Regression methods that flexibly fit hemodynamic response shape – FIR and basis set GLM approaches – were found to eliminate activation-induced FC inflation (without increasing false negatives), whereas alternative methods did not. Consistent with these theoretical results, we found that FIR and basis set approaches significantly reduced task FC estimates in empirical fMRI data" (Cole et al., 2019, p. 12).

While the Reviewer is correct in noting that the FIR approach appears to be preferred by the authors (“We found that the FIR approach reduced task FC estimates the most, consistent with its unique ability to flexibly fit any possible HRF shape, suggesting this as the preferred approach.” (Cole et al., 2019, p. 12)), the presented empirical evidence also supports the effectiveness of the basis set approach.

For example, Figure 4 of the paper demonstrates that both methods perform similarly well in a computational model in terms of false positive rates. Additionally, regarding false negative rates, the basis set approach actually outperforms FIR (Figure 4). Figure 5 further illustrates that both methods yield very similar results regarding the percentage of task-state FC increases.

In Figure 7, FIR is shown to eliminate nearly all differences in FC estimates between the 2-back and the 0-back task FC. The authors interpret this as evidence that FIR is likely the most accurate approach, stating: “we interpret the FIR results as likely being more accurate than the other approaches” (Cole et al., 2019, p. 12). However, this conclusion relies on the assumption that task FC should be identical for the 0-back and 2-back conditions—an assumption that remains uncertain.

Additionally, the FIR approach has some limitations that should be taken into account: The authors themselves state that FIR may be overly flexible, making it more susceptible to overfitting. Moreover, compared to the basis set approach, FIR employs more parameters in the nuisance regression step, leading to a greater loss of temporal degrees of freedom – a drawback that is not explicitly mentioned in the Cole paper.

In summary, we chose the basis set approach over the FIR approach because it demonstrates comparable performance in multiple scenarios and flexibly fits the HRF dynamics, but achieves this performance without losing as many temporal degrees of freedom and potentially reduces the risk of overfitting.

We thank the Reviewer for noting the importance of justifying our choice and added a respective note in the revised manuscript:

Line 679 – 690 (Methods):

*“fMRI scans were obtained using a gradient-echo EPI sequence in two sessions with multi-slice acceleration (*time to repetition* = 720 ms; *time to echo* = 33.1 ms; 2-mm isotropic voxel resolution; flip angle = 52°; multiband acceleration factor = 8)⁵². Both sessions included two 15-minute resting-state scans in opposite phase encoding directions (L/R and R/L) followed by approximately 30 minutes of task fMRI comprising seven tasks (Table 1; each L/R and R/L) that were split between both sessions. fMRI data were downloaded in the minimally preprocessed form. Additionally, 24 head motion parameters, eight mean signals from white matter and cerebrospinal fluid, and four global signals were regressed out according to Parkes et al.⁹² (strategy No. 6). Task-evoked neural activation was removed by simultaneously adding basis-set task regressors. *This method has been shown to be similarly effective as concurrent methods**

but achieves this performance with fewer parameters in the nuisance regression step, preserving more temporal degrees of freedom, and reducing the risk of overfitting¹⁰⁴.

Reviewer comment 2:

This is a bigger picture suggestion. The manuscript uses the terms “more/less cognitively demanding task” in several places. It is specifically invoked to describe the WM and language tasks as more demanding. While I wouldn’t argue with the fact that there are different cognitive demands across these task sets, it is difficult to quantify cognitive demand in a more fine-grained way (especially across these diverse tasks). In fact, coupling is more strongly related to general intelligence for the social and gambling tasks than it is during the language task, which the authors label as more cognitively demanding. The authors do cite a few papers suggesting that the emotion task is less cognitively demanding. It might be useful to either cite other papers when defining high/low cognitive demand, or to try to quantify it some way with the current data (reaction time?). Another approach would be to change the way that the differences in cognitive demand across this set of tasks are framed. I don’t have a perfect solution, but the use of the term along with the reported results led to some confusion on my part as I read the manuscript.

Author’s response 2:

We appreciate the Reviewer’s insightful comment and acknowledge the challenges associated with our classification of tasks into more or less “cognitively demanding”. We recognize that our framing lacked specificity, and a clear explanation was missing in the former version of our manuscript. We thank the Reviewer for bringing this issue to our attention.

To clarify, we define cognitive demand as the extent to which a task can be assumed to require mental effort, analytical reasoning, and problem-solving skills – concepts closely related to g (Spearman, 1904). Based on these considerations, we initially categorized the working memory task, the language task, and the relational processing task as more cognitively demanding, whereas the emotion processing task, the gambling task, the social cognition task and the motor task were considered as less demanding.

However, the Reviewer is absolutely correct in stating that an empirical validation of our conceptual assumptions might be preferable. Thus, we exactly followed their suggestion and conducted a post-hoc analysis testing for relations between g and the performance scores of all tasks (computed in accordance to Greene et al., 2020). This analysis was conducted with all participants in the complete HCP sample for which the required performance data were available (N=737):

Table R1

Associations between in-scanner task performance and general intelligence.

	Task performance r (p)					
	EMO	GAM	LAN	REL	SOC	WM

g-factor	.13 (< .001)	-.07 (.07)	.45 (< .001)	.45 (< .001)	.23 (< .001)	.49 (< .001)
----------	------------------------	------------	------------------------	------------------------	------------------------	------------------------

Note: Pearson correlations (r) depicting the associations between individual task performance and general intelligence scores. The performance scores were computed as outlined in Greene et al. (2020) based on behavioral task measures provided by the Human Connectome Project. No performance scores were available for the motor task. Analyses were corrected for multiple comparisons (Bonferroni corrected threshold for six correlations; sig. $p < .008$) and significant associations are presented in bold. EMO = Emotion processing task; GAM = Gambling task; LAN = Language task; REL = Relational processing task; SOC = Social cognition task; WM = Working memory task.

The results of this post-hoc analysis support our initial classification: The strongest correlations with g ($r \geq .45$) were observed for the language task, the relational processing task and the working memory task, suggesting that these tasks are indeed more cognitively demanding.

We included information about this post hoc analysis in the manuscript to substantiate our classification of cognitive demand. We appreciate the Reviewers comment and think that this analysis strengthens our manuscript:

Line 481 – 490 (Results):

“To enhance conceptual insights, we post-hoc classified the tasks during fMRI recording into more or less cognitively demanding ones. Specifically, we determined the associations (Pearson correlations) between general intelligence and behavioral task performance across all tasks (as operationalized in⁵¹). This analysis was conducted with all HCP participants for whom the respective data were available ($N = 737$). Significant associations (Bonferroni corrected threshold for six correlations; sig. $p < .008$) were observed for the emotion processing task ($r = .13$; $p < .001$), the language task ($r = .45$; $p < .001$), the relational processing task ($r = .45$; $p < .001$), the social cognition task ($r = .23$; $p < .001$), and the working memory task ($r = .49$; $p < .001$). The language task, the relational processing task and the working memory task demonstrated the strongest associations ($r \geq .45$) and were thus considered as tasks of higher cognitive demand.”

Furthermore, we acknowledge that it was not correct to proclaim that prediction accuracy was highest during cognitively demanding tasks, as less cognitively demanding tasks (such as the social cognition task and the gambling task) actually outperformed the tasks we assumed (and now also empirically determined) as cognitively more demanding – in terms of accuracy.

*However, in terms of robustness (whether predictions can be replicated successfully or not), predictions from more cognitively demanding tasks excel. Specifically, those predictions evolved as significant in the internal cross-validation in the main sample **and** in the cross-sample model generalization test (HCP532 \rightarrow HCP232) for resting-state, the language task, the relational processing task, the social cognition task, and the working memory task. Furthermore, predictions from the working memory task and the language task were the only ones that were significant in the internal cross-validation within the main sample **and** within*

*the lockbox sample **and** in the cross-sample model generalization test. We corrected respective paragraphs and thank the Reviewer again for this very valuable notion:*

Abstract:

*“Most **robust** predictions resulted from cognitively demanding tasks and task combinations.”*

Line 501 – 504 (Discussion):

*“On a brain-average level, higher intelligence was only associated with higher coupling during the emotion processing task. However, accounting for region-specific variability in SC-FC coupling enabled significant prediction of individual intelligence scores, with cognitively demanding tasks and coupling information from all tasks **providing strong and most robust (replicable) predictions.**”*

Line 555 – 564 (Discussion):

*“Notably, **strong and most robust (replicable)** prediction performance was achieved with coupling acquired during cognitively demanding tasks (i.e., working memory and the language task). This observation aligns with recent research^{27,49,50,69,71}, and is substantiated by established personality theories, which propose that individual variations in a given personality trait become most apparent during situations relevant for this trait⁷²⁻⁷⁴. Cognitively demanding tasks may rely more on intelligence-relevant neural circuitry, emphasizing meaningful individual differences and thereby improving predictive accuracy **and robustness** (for similar argumentation see⁴⁸). However, prediction from resting state outperformed some of the task conditions. One potential explanation could be that the unconstrained resting state leads more intelligent individuals to follow more complex thoughts, thus unintentionally inducing trait-relevant situations.”*

References

- Cole, M. W., Ito, T., Schultz, D., Mill, R., Chen, R., & Cocuzza, C. (2019). Task activations produce spurious but systematic inflation of task functional connectivity estimates. *NeuroImage*, *189*, 1–18. <https://doi.org/10.1016/j.neuroimage.2018.12.054>
- Cucina, J., & Byle, K. (2017). The Bifactor Model Fits Better Than the Higher-Order Model in More Than 90% of Comparisons for Mental Abilities Test Batteries. *Journal of Intelligence*, *5*(3), Article 3. <https://doi.org/10.3390/jintelligence5030027>
- Deary, I. J., Johnson, W., & Houlihan, L. M. (2009). Genetic foundations of human intelligence. *Human Genetics*, *126*(1), 215–232. <https://doi.org/10.1007/s00439-009-0655-4>
- Dubois, J., Galdi, P., Paul, L. K., & Adolphs, R. (2018). A distributed brain network predicts general intelligence from resting-state human neuroimaging data. *Philosophical Transactions of the Royal Society B: Biological Sciences*, *373*(1756), 20170284. <https://doi.org/10.1098/rstb.2017.0284>
- Gonzalez-Castillo, J., & Bandettini, P. A. (2018). Task-based dynamic functional connectivity: Recent findings and open questions. *NeuroImage*, *180*, 526–533. <https://doi.org/10.1016/j.neuroimage.2017.08.006>
- Greene, A. S., Gao, S., Noble, S., Scheinost, D., & Constable, R. T. (2020). How Tasks Change Whole-Brain Functional Organization to Reveal Brain-Phenotype Relationships. *Cell Reports*, *32*(8), 108066. <https://doi.org/10.1016/j.celrep.2020.108066>
- Haier, R. J. (2023, Juli). *The Neuroscience of Intelligence*. Cambridge Core; Cambridge University Press. <https://doi.org/10.1017/9781009295055>
- Kan, K.-J., Psychogiopoulos, A., Groot, L. J., de Jonge, H., & ten Hove, D. (2024). Why Do Bi-Factor Models Outperform Higher-Order g Factor Models? A Network Perspective. *Journal of Intelligence*, *12*(2), Article 2. <https://doi.org/10.3390/jintelligence12020018>
- Kopetzky, S. J., Li, Y., Kaiser, M., Butz-Ostendorf, M., & Alzheimer's Disease Neuroimaging Initiative. (2024). Predictability of intelligence and age from structural connectomes. *PloS One*, *19*(4), e0301599. <https://doi.org/10.1371/journal.pone.0301599>
- Osmanlioglu, Y., Alappatt, J. A., Parker, D., & Verma, R. (2020). Analysis of Consistency in Structural and Functional Connectivity of Human Brain. *2020 IEEE 17th International Symposium on Biomedical Imaging (ISBI)*, 1694–1697. <https://doi.org/10.1109/ISBI45749.2020.9098412>
- Popp, J. L., Thiele, J. A., Faskowitz, J., Seguin, C., Sporns, O., & Hilger, K. (2024). Structural-functional brain network coupling predicts human cognitive ability. *NeuroImage*, *290*, 120563. <https://doi.org/10.1016/j.neuroimage.2024.120563>

- Spearman, C. (1904). *General intelligence, 'objectively determined and measured. First published in American Journal of Psychology, 15, 201-293.*
- Thiele, J. A., Faskowitz, J., Sporns, O., & Hilger, K. (2022). Multitask Brain Network Reconfiguration Is Inversely Associated with Human Intelligence. *Cerebral Cortex*, bhab473. <https://doi.org/10.1093/cercor/bhab473>
- Thiele, J. A., Faskowitz, J., Sporns, O., & Hilger, K. (2024). Choosing explanation over performance: Insights from machine learning-based prediction of human intelligence from brain connectivity. *PNAS Nexus*, 3(12), pgae519. <https://doi.org/10.1093/pnasnexus/pgae519>
- Zhang, Z., Allen, G. I., Zhu, H., & Dunson, D. (2019). Tensor network factorizations: Relationships between brain structural connectomes and traits. *NeuroImage*, 197, 330–343. <https://doi.org/10.1016/j.neuroimage.2019.04.027>